# JiuZhang3.0: Efficiently Improving Mathematical Reasoning by Training Small Data Synthesis Models

**Kun Zhou**[1][*]**, Beichen Zhang**[2][*]**, Jiapeng Wang**[2]**, Zhipeng Chen**[2]**, Wayne Xin Zhao**[2][†]**,**
**Jing Sha**[3]**, Zhichao Sheng**[3]**, Shijin Wang**[3,4][†]**, Ji-Rong Wen**[2]
[1]School of Information, Renmin University of China
[2]Gaoling School of Artificial Intelligence, Renmin University of China
[3]iFLYTEK Research    [4]iFLYTEK AI Research (Central China)
`francis_kun_zhou@163.com, {zhangbeichen724,batmanfly}@gmail.com,`
`{wangjp1010,zhipeng_chen,jrwen}@ruc.edu.cn, {jingsha,sjwang3}@iflytek.com`

## Abstract

Mathematical reasoning is an important capability of large language models (LLMs) for real-world applications. To enhance this capability, existing work either collects large-scale math-related texts for pre-training, or relies on stronger LLMs (*e.g.,* GPT-4) to synthesize massive math problems. Both types of work generally lead to large costs in training or synthesis. To reduce the cost, based on open-source available texts, we propose an efficient way that trains a small LLM for math problem synthesis, to efficiently generate sufficient high-quality pre-training data. To achieve it, we create a dataset using GPT-4 to distill its data synthesis capability into the small LLM. Concretely, we craft a set of prompts based on human education stages to guide GPT-4, to synthesize problems covering diverse math knowledge and difficulty levels. Besides, we adopt the gradient-based influence estimation method to select the most valuable math-related texts. The both are fed into GPT-4 for creating the knowledge distillation dataset to train the small LLM. We leverage it to synthesize 6 million math problems for pre-training our JiuZhang3.0 model. The whole process only needs to invoke GPT-4 API 9.3k times and use 4.6B data for training. Experimental results have shown that JiuZhang3.0 achieves state-of-the-art performance on several mathematical reasoning datasets, under both natural language reasoning and tool manipulation settings. Our code and data will be publicly released in `https://github.com/RUCAIBox/JiuZhang3.0`.

## 1 Introduction

Large language models (LLMs) have shown remarkable capabilities on a variety of tasks [1–3]. However, they still struggle in solving complex mathematical problems [4]. Recent work has shown that it is an effective approach to training LLMs on math-related data for improving the mathematical reasoning ability [5, 6]. Typically, they either collect the math-related data from the available corpora (*e.g.,* webpages and books) for pre-training [7–9], or rely on stronger LLMs to synthesize high-quality math problems for fine-tuning [10–12]. Despite the success, existing approaches would generally cause large training or inference costs. Due to the complexity and diversity of mathematical problems, the former type of work mostly needs to collect a large-scale corpus (*e.g.,* 120B data for Deepseek-Math) for training, which greatly increases the training cost [5, 7, 8]. Similarly, to guarantee the knowledge coverage and effectiveness of the synthetic problems, the latter type of work relies on

---

[*]Equal contributions.
[†]Corresponding authors.

38th Conference on Neural Information Processing Systems (NeurIPS 2024).

stronger LLMs with larger scales (*e.g.,* GPT-4) to create massive math problems, leading to larger inference cost [6, 10, 11]. In Figure 1, we show our estimated total costs of re-implementing two math-related LLMs (>$40000), details are in Appendix A

In this work, we aim to develop a relatively low-cost data synthesis approach for improving the mathematical reasoning abilities of LLMs. Our key idea is that **the data synthesis capability can be well learned by small LLMs**. Here, *small* is a relative wording, which is in contrast with the extremely large or costly data synthesis models used in prior studies [6, 13], such as GPT-4 or Qwen-72B. Actually, existing work [10, 14, 15] has extensive evidence of strong learning and adaptation abilities of small LLMs for new tasks and domains with suitable strategies (*e.g.,* training with high-quality supervised data), including math, science, and complex multimodal tasks. However, this exploration has been neglected in prior efforts on data

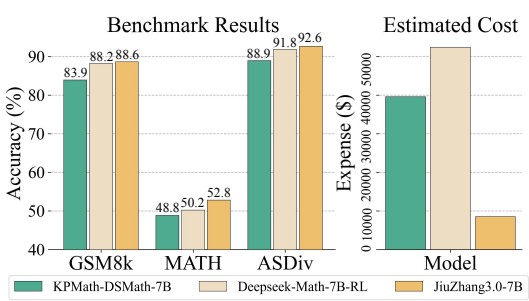

Figure 1: The comparison of existing work and our method in task performance and the total cost.

synthesis. This attempt can be essentially generalized to a broader problem: *whether a small (or weak) model can produce high-quality data that is useful for training a large (or strong) model?* Inspired by this motivation, we seek to train a relatively small yet powerful LLM for synthesizing high-quality math-related data.

However, due to the diverse and complex nature of math problems, it is challenging to train well-performing small LLMs for synthesizing high-quality ones. Although we can leverage GPT-4, it is not efficient to use it for synthesizing a large-scale knowledge distillation (KD) dataset. Specifically, we aim to build the KD dataset through a low-cost strategy, but it can sufficiently capture diverse and useful knowledge about math problem synthesis. Thus, we should guarantee the knowledge coverage and usefulness of the instances within the KD dataset. To achieve it, we first craft a set of prompts, and each prompt corresponds to an education stage of humans, *e.g.,* middle school and college. Using the above prompts, the instances within the dataset can well cover broad mathematical knowledge and different difficulty levels. Besides, for usefulness, we estimate the influence of the available math-related texts and downstream tasks, by computing the gradient similarity between their corresponding synthetic data and the task instances. Then, we select the top-ranking math-related texts into the KD dataset, which are high-value ones with more positive influence on downstream tasks. By feeding the texts with prompts into GPT-4, we collect the outputs to build the KD dataset.

Based on the KD dataset, we train DeepSeekMath-7B as our data synthesis model, which is much smaller than other commonly-used LLMs in existing work [6, 12, 13], *e.g.,* GPT-4 and Qwen-72B. Owing to our data selection strategy, we only require GPT-4 to generate 9,335 instances based on the selected most valuable texts for training it. Then, we utilize the crafted prompts to guide it for synthesizing high-quality problems. Benefiting from the strong data synthesis capability of the small model, we only need to synthesize 5,984,584 high-quality math problems (4.6B tokens) for pre-training our JiuZhang3.0. Thus, the total inference and training cost is much less than existing work, as shown in Figure 1. After pre-training, we also collect open-source math instructions to fine-tune JiuZhang3.0. The experimental results have shown that JiuZhang3.0 can mostly outperform state-of-the-art methods across 18 evaluation datasets, in both the natural language reasoning and tool manipulation settings. Our contributions are summarized as follows:

(1) our research provides compelling evidence that it is feasible to efficiently train a small LLM (7B) for synthesizing training data to improve the mathematical reasoning of LLMs. As results shown in Section 4.3, its synthetic data is more useful than larger LLMs in improving the performance.

(2) we propose an efficient solution for training LLMs to improve mathematical reasoning, which only needs to invoke GPT-4 API 9.3k times and pre-train on 4.6B high-quality synthetic data, with nearly 20% total cost of existing state-of-the-art methods.

(3) JiuZhang3.0 achieves state-of-the-art performance among open-source LLMs on several tasks and settings, *e.g.,* 52.8 (JiuZhang3.0-7B) vs. 50.2 (DeepSeekMath-7B-RL) on MATH, 89.8 (JiuZhang3.0-8×7B) vs. 86.4 (MAmmoTH2-8×7B-Plus) on GSM8k in the natural language reasoning setting.

## 2 Related Work

**Large Language Models.** LLMs have demonstrated remarkable capabilities in a variety of NLP tasks, and commercial LLMs like ChatGPT, Claude, and Gemini [2, 16, 17], represent cutting-edge capabilities. Meanwhile, the performance of open-source models (*e.g.,* LLaMA-3, Mixtral) has also developed rapidly [18, 19]. To further improve the capability of LLMs on special tasks or domains, existing work mainly focuses on optimizing the following aspects: (1) prompt engineerings such as chain-of-thought and tree-of-thought [20, 21]; (2) continual pre-training on a domain-specific or task-specific corpus, improving the model to deal with downstream tasks [5, 7, 8, 22, 23]; (3) supervised fine-tuning, which involves fine-tuning the model on related instruction datasets, enhancing LLMs to follow special task instructions [24, 25]; (4) other strategies including RLHF [26], tool augmentation [27], decoding optimization [28] and *et al.* We aim to efficiently improve the capability of LLMs for mathematical reasoning by pre-training on synthetic data.

**Mathematical Reasoning.** Despite the impressive progress, mathematical reasoning remains a weak aspect of LLMs. To enhance LLMs' ability in mathematical reasoning, researchers have proposed a surge of methods from the aspects of prompting, pre-training and fine-tuning. For prompting, the chain-of-thought (CoT) prompts have been widely used to guide LLMs for performing multi-step reasoning on complex math problems [20]. Based on it, following work utilizes tools [27, 29–33] and verifiers [9, 34–36], to further improve the accuracy of the mathematical reasoning process. For pre-training, existing work [5, 7, 9, 37, 38] collects a large-scale math-related corpus and continually pre-training open-source LLMs on it. Supervised fine-tuning methods focus on using relatively less high-quality data for training the LLM, which are typically math-related instructions [10, 12, 39]. Recent studies show that the complexity of mathematical reasoning demands high-quality instruction pairs, leading to reliance on advanced LLMs like GPT-4 for data synthesis [6, 40, 41]. The pre-training and fine-tuning methods generally lead to large training and data annotation costs, respectively. Our work aims to train a small LLM specially for math problem synthesis, which can efficiently produce sufficient data for training.

**Data Synthesis.** For complex tasks and scenarios (*e.g.,* mathematical reasoning), it is necessary to collect a substantial amount of data for training the LLM to enhance it. However, the available data may not be sufficient, hence researchers have explored using automatically synthetic data with consistent distribution to real data, to enrich the training corpus [42–48]. For data synthesis on mathematical reasoning tasks, existing work can be roughly categorized into the following two types, according to their based guided information. The first type of work starts with existing problems or math-related texts to synthesize similar problems or solutions [10, 40, 41, 49, 50]. The other type of work relies on available knowledge points, and devises special prompts to guide LLMs for synthesizing related problems with the solutions [6, 51]. As correctness is important, the two types of work generally design rules to check and remove wrong ones. In this work, based on the data synthesis model, we also construct the multi-source math corpora, and craft several prompts to guide it in producing diverse and useful math problems.

## 3 Approach

In this section, we present our approach that aims to train a small LLM for synthesizing math problems. First, we initialize the data synthesis model by training it on the KD dataset, composed of crafted prompts, randomly sampled math-related texts, and the corresponding synthetic problems and solutions from GPT-4. Then, we improve its data synthesis capability by retraining it on the updated knowledge distillation dataset, where we add the high-value math-related texts selected by gradient-based influence estimation strategy. Finally, we utilize the model to synthesize massive high-quality math problems for training JiuZhang3.0, based on the multi-source math-related corpus.

### 3.1 Preliminary

We focus on training a small data synthesis LLM, for synthesizing high-quality math problem-solution pairs to pre-train LLMs and improve its mathematical reasoning capability. To guarantee the quality of the synthetic data, we utilize GPT-4 to create the knowledge distillation (KD) dataset $\mathcal{D}_{KD} = \{p_i, t_i, \hat{q}_i, \hat{s}_i\}_{i=1}^N$ for training the small LLM, where the math-related text $t_i$ and the prompt

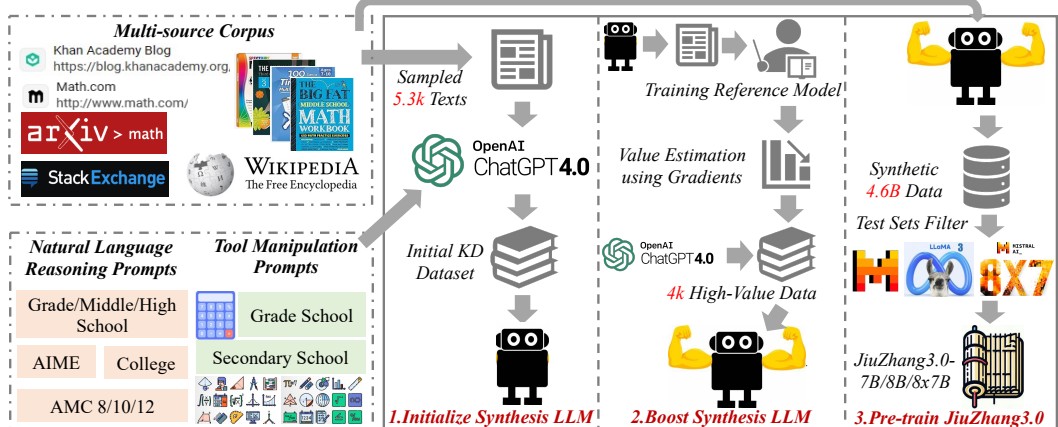

Figure 2: The pipeline of our approach. We first initialize the data synthesis LLM by distilling the knowledge from GPT-4 on randomly sampled data, then boost it using the high-value data selected by gradient-based value estimation strategy, finally utilize it for synthesizing data to train JiuZhang3.0.

$p_i$ are the input of GPT-4, $\hat{q}_i$ and $\hat{s}_i$ are its synthetic math problem and solution respectively. Then, we train the small LLM on $\mathcal{D}_{KD}$ to imitate the data synthesis ability of GPT-4. Finally, we leverage it to synthesize the pre-training dataset $\mathcal{D} = \{q_i, s_i\}_{i=1}^{M}$ based on all the collected math-related texts with randomly selected prompts $\{p_i, t_i\}_{i=1}^{M}$, which are used for training our JiuZhang3.0.

## 3.2 Initializing Data Synthesis Model

In this work, we consider the natural language reasoning and tool manipulation settings, where LLMs require solving the problem by generating a natural language solution [20] and an executable program with external interpreters [52], respectively. Thus, we train the data synthesis LLMs on the initial KD dataset, containing special prompts, math-related texts, and GPT-4 outputs for the two settings.

### 3.2.1 Prompts for Math Problem Synthesis

We aim to craft a prompt set that can well cover the knowledge points and difficulty levels in human math education. Thus, for the natural language reasoning and tool manipulation settings, we manually craft prompt templates respectively, and each corresponds to a certain education stage.

**Prompts for Natural Language Reasoning.** We consider the following 4 human education stages and 4 worldwide competitions, *i.e.,* Grade School, Middle School, High School, and College; AMC (American Mathematics Competition) 8, AMC 10, AMC 12, and AIME (American Invitational Mathematics Examination). Based on these, we design 8 prompts with corresponding instructions and guidelines. We show an example of grade school math problem synthesis:

*## Instruction: Create an age-appropriate math word problem for grade school students based on the provided math content.*

*## Guidelines: [Problem]: Craft a concise math word problem suitable for grade school, focusing on basic arithmetic operations, number sense, simple shapes, ... [Solution]: Provide a clear, step-by-step solution to the problem using simple language that a grade school student could understand, ...*

**Prompts for Tool Manipulation.** We consider 2 types of math problems,*i.e.,* Grade School and Secondary School Competitions, as the competition math problems may need tools for advanced math operations (*e.g.,* integral computation), while grade school math problems are much easier and can be solved by basic operations (*e.g., +-\*/*). As tool manipulation solves the problem via executable programs, we use more words in prompts to emphasize the data format and show an example as follows:

*## Instruction: Please gain inspiration from the following random math content to create a high-quality ... Present your output in two distinct sections: [Problem Description] and [Solution].*

*## **Guidelines:** [Problem]: This should be completely self-contained, providing all the contextual information one needs to understand, ... [Solution]: Offer a comprehensive, correct solution that accurately addresses the [Problem] you provided using Python code, ...*

*## **Example:** [Problem Description] Janet buys 3 pounds of broccoli, ... [/Problem Description] [Solution] def spending(): cost = 4 ...[/Solution]*

### 3.2.2  Knowledge Distillation from GPT-4

Based on the prompts, we build the KD dataset to train our data synthesis LLM. We randomly sample 5,336 math-related texts, and concatenate each one with a randomly selected prompt, to compose the input *i.e.,* $[p_i; w_i]$. Then, we feed it into GPT-4, and extract the synthetic math problem $\hat{q}_i$ and solution $\hat{s}_i$ from its output using regular expressions, to compose the KD dataset $\mathcal{D}_{KD} = \{p_i, w_i, \hat{q}_i, \hat{s}_i\}_{i=1}^{N_{ini}}$. Next, we utilize it to train the synthesis model, and the learning objective is:

$$L(\theta_{syn}) = \sum_{i=1}^{N_{ini}} \log P([\hat{q}_i; \hat{s}_i]|[p_i; w_i]), \tag{1}$$

where $\theta_{syn}$ denotes the parameters of the data synthesis LLM. In this way, we can teach it to generate new math problem-solution pairs based on prompts and math-related texts by imitating GPT-4.

### 3.3  Boosting Synthesis Model using High-Value Data

After initialization, we further improve the data synthesis LLM using high-value KD data. However, it would be costly if we first utilize GPT-4 to generate candidates and then select high-value ones. Therefore, we propose an efficient way that leverages the data synthesis LLM for generating candidates, and then selects valuable ones to feed into GPT-4. Specifically, we incorporate the gradient-based method [53] to estimate the influence of each synthetic instance for downstream math-related tasks, and select the top-ranking ones to update the KD dataset for retraining the data synthesis LLM.

### 3.3.1  Gradient-based Data Value Estimation

According to the influence formulation [54], at a certain training step of a model parameterized by $\theta$, the influence of a training instance $z$ on another instance $z'$ can be estimated by computing the similarity between their produced gradients, denoted as:

$$\text{Inf}(z, z') \propto Sim(\nabla l(z, \theta), \nabla l(z', \theta)). \tag{2}$$

By using it, we can measure the value of each synthetic data by computing its gradient similarity with downstream math-related task data. Concretely, we first train a reference model using LoRA, then compute its gradients on LoRA parameters as the features, to help estimate the data value.

**Training Reference Model using LoRA.**  Inspired by existing work [53], we train a LLM for mathematical problem solving using LoRA [55] as the reference model. As LoRA only requires to optimize the low-rank adapters in the LLM, we can efficiently train the reference model on limited computation resources, and reduce the number of trainable parameters for efficient computation of gradient similarity. Besides, to further reduce the training cost, we randomly select a subset of synthetic math problems generated by the data synthesis model, denoted as $\mathcal{D}_{lora} = \{q_i, s_i\}_{i=1}^{M_l}$. Then, we train the reference model to predict the solution based on the given problem, denoted as:

$$L_{ref}(\theta_{lora}) = \sum_{i=1}^{M_l} \log P(s_i|q_i) \tag{3}$$

where $\theta_{lora}$ denotes the parameters of LoRA, and $M_l$ is the number of training data.

**Computing Gradient Features.**  After training the reference model, we compute the gradients of LoRA parameters as the feature of each synthetic instance. As their dimension is large, we follow existing work [56] that performs random projection to obtain the low-dimensional features as:

$$\hat{\nabla} l_{ref}(z, \theta_{lora}) = \Pi^\top \nabla l_{ref}(z, \theta_{lora}), \tag{4}$$

where $z = \langle p_i, w_i, q_i, s_i \rangle$ denotes a synthetic instance, $\Pi \in \mathbb{R}^{d' \times d}$ is a projection matrix initialized by the Rademacher distribution, its entries are -1 or 1, $d'$ and $d$ are the dimensions before and after projection, respectively. According to the Johnson-Lindenstrauss Lemmas [57], this operation can nearly preserve the gradient distances, ensuring the usefulness of the low-dimensional features.

**Estimating Data Value.** By using Eq. 4, we can compute the gradient features for synthetic instances. Then, we randomly sample $M_D$ instances from the training sets of downstream math-related datasets $\{z'_i\}_{i=1}^{M_D}$, where $z'_i = \langle \tilde{q}_i, \tilde{s}_i \rangle$, and also compute their gradient features. Next, we estimate the value of each synthetic instance by computing the similarity between its gradient feature and the average feature of all the sampled downstream instances as:

$$V(z) = \texttt{Cosine}\big(\hat{\nabla}l_{ref}(z, \theta_{lora}), \frac{1}{M_D} \sum_{i=1}^{M_D} \hat{\nabla}l_{ref}(z'_i, \theta_{lora})\big), \tag{5}$$

where $\texttt{Cosine}(x, y)$ computes the cosine similarity between the two vectors. In this way, the instance with higher data value would lead to a more positive influence on the downstream math-related tasks.

### 3.3.2 Retraining Data Synthesis Model

Based on the estimated values, we can rank all the synthetic instances, and the top-ranking $N_{add}$ ones can be regarded as the most valuable data $\{\langle p_i, w_i, q_i, s_i \rangle\}_{i=1}^{N_{add}}$ for improving downstream math-related tasks. Thus, we utilize GPT-4 to regenerate the synthetic math problems based on their prompts and original math-related texts, to acquire corresponding more high-quality math problems and solutions. Then, we add the new GPT-4 synthetic data $\{p_i, w_i, \hat{q}_i, \hat{s}_i\}_{i=1}^{N_{add}}$ into the KD dataset, and the new data is capable of guiding the small LLM to generate more useful math problems for downstream tasks. Next, we retrain the data synthesis LLM with the updated KD dataset using Eq. 1.

### 3.4 Pre-training JiuZhang3.0 using Synthetic Data

After training the data synthesis LLM, we construct the multi-source corpus containing rich math-related texts to cover more knowledge and scenarios. Then, we synthesize massive math problems based on it, which are used for pre-training JiuZhang3.0.

**Constructing Multi-source Corpus.** We consider the following data types and select the corresponding open-source datasets to compose the math-related multi-source corpus.

• *Webpages:* we use the OpenWebText corpus [58], which consists of 6.3M math-related web documents extracted from Common Crawl.

• *Books:* we use the Mathpile-textbook dataset [59], including 4K educational textbooks, lecture notes and synthetic books.

• *Papers:* we use the Mathpile-Arxiv dataset [59], and select the high-quality ones according to the estimated scores (0.6-0.9), which are released by AutoMathText [60].

• *QA Data:* we select the StackExchange subset of the MMIQC dataset [41], which contains 1.2M processed real-world math question-answering pairs.

• *Wikipedia:* we use the Mathpile-Wikipedia dataset [59], consisting of 106K documents from math-related entries in Wikipedia.

**Data Synthesis for Training JiuZhang3.0.** For each instance within the multi-source corpus, we randomly select a prompt from the prompt set and embed the text into the prompt to compose the input. Then, we feed inputs into the data synthesis model, to generate the math problems and solutions for composing the synthesis dataset $\mathcal{D} = \{q_i, s_i\}_{i=1}^{M}$. Here, we follow existing work [5, 13] to filter out the instances with 10-grams overlap to both inputs and outputs from test sets of downstream evaluation tasks. We synthesize about 6M math problems (4.6B tokens) in total, which are used for pre-training JiuZhang3.0 to predict the solution based on the given problems.

Table 1: Results on 6 datasets in the natural language reasoning setting. The best and second-best ones among LLMs with similar scales are marked in bold and underlined respectively.

| Models | GSM8k | MATH | SVAMP | ASDiv | MAWPS | CARP | Avg. |
|---|---|---|---|---|---|---|---|
| ChatGPT | 76.6 | 38.2 | 83.7 | 87.7 | 96.9 | 41.3 | 70.7 |
| GPT-4 | 92.2 | 65.4 | 92.9 | 94.3 | 96.6 | 53.6 | 82.5 |
| Qwen-1.5-110B | 85.4 | _49.4_ | 86.2 | 85.1 | 94.3 | **53.6** | 75.7 |
| Qwen-1.5-72B | 77.6 | 39.4 | 83.1 | 85.1 | 95.8 | _53.0_ | 72.3 |
| Mixtral-8×7B | 74.4 | 29.0 | 76.5 | 78.5 | 93.9 | 38.8 | 65.2 |
| Llemma-34B | 60.2 | 24.6 | 68.0 | 75.6 | 89.8 | 36.5 | 59.1 |
| Intern-Math-20B | 64.9 | 27.4 | 74.9 | 79.6 | 94.4 | 42.3 | 63.9 |
| ChatGLM-Math-32B | 82.6 | 40.6 | - | - | - | - | - |
| MAmmoTH2-8x7B-Plus | _86.4_ | 47.0 | _90.0_ | _92.2_ | **97.0** | 45.8 | _76.4_ |
| JiuZhang3.0-8x7B (Ours) | **89.8** | **53.8** | **90.2** | **93.1** | _96.7_ | 52.3 | **79.3** |
| DeepSeek-7B | 13.6 | 4.8 | 40.8 | 52.1 | 65.4 | 10.3 | 31.2 |
| Mistral-7B | 41.2 | 13.6 | 64.7 | 68.5 | 87.5 | 14.9 | 48.4 |
| LLaMA-3-8B | 54.5 | 19.6 | 68.5 | 72.8 | 90.5 | 29.2 | 55.9 |
| Gemma-7B | 54.1 | 19.6 | 69.7 | 74.2 | 89.0 | 30.5 | 56.2 |
| Qwen-1.5-7B | 60.5 | 28.2 | 64.9 | 74.9 | 90.1 | 38.6 | 59.5 |
| Llemma-7B | 39.2 | 18.4 | 56.9 | 69.0 | 82.7 | 31.8 | 49.7 |
| InternLM-Math-7B | 45.9 | 15.8 | 67.3 | 71.2 | 88.3 | 28.0 | 52.8 |
| Rho-1-Math-7B | 66.3 | 31.0 | 78.5 | 79.2 | 94.0 | 36.7 | 64.3 |
| DeepSeekMath-7B | 64.1 | 34.2 | 73.7 | 82.7 | 92.7 | 44.4 | 65.3 |
| Mistral-7B-MMIQC | 75.0 | 34.2 | 73.5 | 82.1 | 90.1 | 36.5 | 65.2 |
| MetaMath-Mistral-7B | 77.8 | 29.6 | 79.6 | 81.2 | 93.7 | 30.5 | 65.4 |
| Abel-7B-002 | 80.4 | 29.6 | 78.8 | 82.7 | 93.5 | 33.2 | 66.4 |
| WizardMath-7B-1.1 | 82.2 | 32.8 | 80.7 | 84.2 | 93.8 | 31.9 | 67.6 |
| Math-Shepherd-Mistral-7B | 84.3 | 34.4 | 82.9 | 82.8 | 92.5 | 32.9 | 68.3 |
| KPMath-DSMath-7B | 83.9 | 48.8 | 81.5 | 88.9 | 94.8 | - | - |
| MAmmoTH2-7B-Plus | 84.2 | 46.2 | _90.3_ | 90.3 | 95.8 | 44.3 | 75.2 |
| MAmmoTH2-8B-Plus | 84.4 | 41.2 | 89.9 | 89.9 | _97.1_ | 44.8 | 74.6 |
| DeepSeekMath-7B-Instruct | 82.3 | 45.8 | 83.7 | 90.1 | 95.7 | 45.8 | 73.9 |
| DeepSeekMath-7B-RL | 88.2 | 50.2 | 87.3 | 91.8 | 95.5 | **51.6** | 77.4 |
| JiuZhang3.0-7B (Ours) | **88.6** | **52.8** | **90.4** | **92.6** | **97.3** | _51.0_ | **78.8** |
| JiuZhang3.0-8B (Ours) | **88.6** | _51.0_ | 89.4 | **92.6** | _97.1_ | 50.9 | _78.3_ |

# 4 Experiments

## 4.1 Experimental Settings

For our JiuZhang3.0, we follow existing work [13] that train the 7B, 8B and 8×7B versions based on Mistral-7B [61], LLaMA-3-8B [18], and Mixtral-8×7B [19]. During training, we first pre-train it on our synthetic 4.6B math problem-solution pairs and then fine-tune it on the collected multiple open-source instruction datasets. We evaluate JiuZhang3.0 in two settings, *i.e.,* natural language reasoning and tool manipulation. More details about the fine-tuning data, evaluation datasets, baseline methods, and implementation details are in Appendix B, C, D and E, respectively.

## 4.2 Results and Analysis

**Natural Language Reasoning.** The results of this setting are shown in Table 1. First, the baseline methods trained on math-related data perform better than others. Among them, DeepSeekMath-7B is the best-performed base LLM, and DeepSeekMath-7B-RL also performs better than other baselines, since they have been pre-trained on 120B corpus containing rich math-related data. Besides, KPMath-DSMath-7B and MAmmoTH2 also perform well. Concretely, KPMath-DSMath-7B is trained on nearly 1M synthetic math problems produced by GPT-4, and MAmmoTH2 also utilizes the GPT-4, Mixtral-8×7B, and Qwen-72B to extract and refine the problems existing in the webpages. The acquired problems can greatly improve their performance in math problem solving. In our approach, we also utilize synthetic math problems to train our JiuZhang3.0-7B and 8B models. Differently,

Table 2: Results on 5 other datasets with different data formats or related to interdisciplinary fields, and we abbreviate MMLU-STEM into M-STEM. The best and second-best methods among LLMs with similar scales are marked in bold and underlined respectively.

| Models | TabMWP | AQuA | SAT-Math | M-STEM | OCW-Math | Avg. |
|---|---|---|---|---|---|---|
| ChatGPT | 82.0 | 53.9 | 78.1 | 63.5 | 11.0 | 57.7 |
| GPT-4 | 90.8 | 76.9 | 96.9 | 77.1 | 26.5 | 73.6 |
| Qwen-1.5-110B | 80.5 | 64.6 | **87.5** | 71.5 | 14.0 | 63.6 |
| Qwen-1.5-72B | 56.1 | 55.1 | **87.5** | 68.8 | 7.7 | 55.0 |
| Mixtral-8×7B | 67.3 | 48.0 | 65.6 | 62.3 | 8.8 | 50.4 |
| Llemma-34B | 57.1 | 46.1 | 71.9 | 54.3 | 11.8 | 48.2 |
| Intern-Math-20B | 63.4 | 44.1 | 65.6 | 62.3 | 7.0 | 48.5 |
| MAmmoTH2-8x7B-Plus | 62.7 | 55.9 | 81.2 | **71.8** | 18.8 | 58.1 |
| JiuZhang3.0-8x7B (Ours) | **84.7** | **65.4** | 81.2 | 66.9 | **23.5** | **64.3** |
| Mistral-7B | 37.3 | 34.3 | 56.2 | 49.5 | 3.3 | 36.1 |
| LLaMA-3-8B | 67.5 | 46.5 | 56.2 | 54.4 | 7.7 | 46.5 |
| Gemma-7B | 60.9 | 42.9 | 71.9 | 57.7 | 4.8 | 47.6 |
| Llemma-7B | 49.2 | 37.8 | 62.5 | 45.8 | 7.7 | 40.6 |
| Rho-1-Math-7B | 55.5 | 49.2 | 75.0 | 54.9 | 6.2 | 48.2 |
| DeepSeekMath-7B | 69.8 | 51.6 | 84.4 | 56.1 | 17.6 | 55.9 |
| DeepSeekMath-7B-Instruct | 70.5 | 60.6 | 84.4 | 57.9 | 19.5 | 58.6 |
| MAmmoTH2-7B-Plus | 54.7 | **62.2** | 84.4 | 64.0 | 15.1 | 56.1 |
| MAmmoTH2-8B-Plus | 75.1 | 57.5 | **87.5** | 65.7 | 14.7 | 60.1 |
| JiuZhang3.0-7B (Ours) | 74.8 | 59.4 | 81.2 | 53.6 | 20.2 | 57.8 |
| JiuZhang3.0-8B (Ours) | **79.2** | **62.2** | 84.4 | 60.4 | **21.3** | **61.5** |

Table 3: Results on 6 mathematical reasoning datasets under the tool manipulation setting. The best and second-best methods are marked in bold and underlined respectively.

| Models | GSM8k | MATH | G-Hard | SVAMP | TabMWP | ASDiv | MAWPS | Avg. |
|---|---|---|---|---|---|---|---|---|
| ChatGPT (PAL) | 78.6 | 38.7 | 67.6 | 77.8 | 79.9 | 81.0 | 89.4 | 73.3 |
| GPT-4 (PAL) | 97.0 | 69.7 | 77.6 | 94.8 | 95.9 | 92.6 | 97.7 | 89.3 |
| CodeLLama | 34.0 | 16.6 | 33.6 | 59.0 | 61.4 | 79.6 | - | - |
| MAmmoTH-7B-Mistral | 75.0 | 40.0 | - | - | - | - | - | |
| MathCoder-7B-CL | 67.8 | 30.2 | - | 70.7 | - | - | - | |
| ToRA-7B-Code | 72.6 | 44.6 | 56.0 | 70.4 | 51.6 | 78.7 | 91.3 | 66.5 |
| MARIO-OVM-7B | 74.5 | 47.7 | - | - | - | - | - | - |
| MMOS-CODE-7B | 73.9 | 44.3 | - | 76.4 | - | 78.6 | - | - |
| OpenMath-Mistral-7B | 80.2 | 44.5 | 63.7 | 82.4 | 70.0 | 82.7 | 95.4 | 74.1 |
| Rho-1-Math-7B-Code | 81.3 | 51.8 | 63.1 | 80.8 | 70.1 | 85.5 | 94.5 | 75.3 |
| JiuZhang3.0-7B (Ours) | 82.4 | 53.0 | **64.9** | 89.2 | 75.6 | **88.3** | 96.6 | 78.6 |
| JiuZhang3.0-8B (Ours) | **82.9** | **53.4** | 64.4 | 89.2 | 79.9 | 87.5 | **97.3** | **79.2** |

our used synthesis model is a much smaller 7B LLM, which has been trained by distilling the data synthesis capability from GPT-4. Thus, it can guarantee the quality of the synthetic data, and helps JiuZhang3.0 models perform the best across most of the dataset. The higher quality also reduces the data amount requirement for pre-training. Owing to our designed high-value data selection strategy, we can also reduce the times of invoking GPT-4 API for knowledge distillation. As noted in Figure 1, the total cost of our approach is nearly only 20% of the compared baselines, indicating its efficiency.

The results of other datasets with different data formats or related to other fields are shown in Table 2. As the listed datasets focus on evaluating the different aspects, the performance of LLMs also differ a lot. For TabMWP, AQuA, and OCW-Math, our JiuZhang3.0-8B and JiuZhang3.0-8×7B achieve the best performance. The three datasets require the understanding of table data, algebra, and undergraduate-level science knowledge respectively, which may have been covered in our synthetic math problems guided by the multi-source corpus. However, our JiuZhang3.0 models perform not well on MMLU-STEM. It indicates the shortcoming of our approach that our prompts and math-related texts might not well cover the knowledge from other subjects.

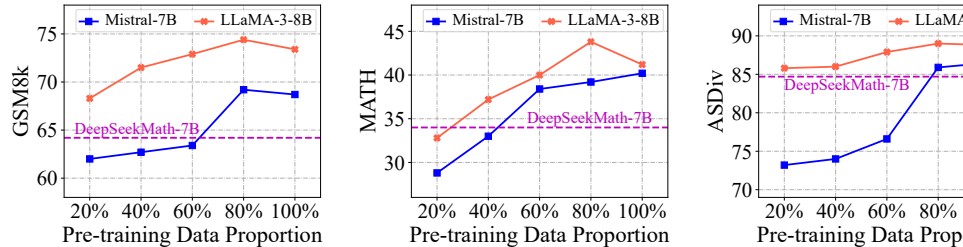

Figure 3: Performance changes with the increasing of the pre-training data proportion for our approach. We also show the best-performed base LLM DeepSeekMath-7B using dashed line.

Table 4: Ablation and variation studies in the natural language reasoning setting. We randomly sample 100k synthetic data and 50k instruction data for efficient test.

| Variation | Models | GSM8k | MATH | ASDiv | CARP |
|---|---|---|---|---|---|
| - | Ours | 78.6 | **32.8** | **84.5** | 36.2 |
| Ablation | w/o Prompt Set | 76.9 | 27.8 | 81.4 | 34.5 |
| | w/o Math-related Texts | 76.4 | 28.6 | 83.8 | 31.9 |
| | w/o Boosting Retraining | 77.9 | 31.6 | 83.8 | 34.7 |
| | w/o Value Estimation | 78.8 | 31.0 | 83.4 | 34.3 |
| | w/o using GPT-4 for Boosting | 79.0 | 28.4 | 82.9 | **36.3** |
| Synthesis LLM | - ChatGPT | 77.0 | 26.6 | 83.0 | 34.3 |
| | - Mixtral-8×7B | 77.6 | 26.8 | 82.9 | 33.1 |
| | - DeepSeekMath-RL-7B | 77.1 | 27.2 | 82.5 | 32.9 |
| | - LLaMA-3-8B-Instruct | 75.7 | 26.2 | 81.5 | 31.4 |
| Data Selection | - Random Sampling | 78.8 | 31.0 | 83.4 | 34.3 |
| | - Perplexity | 77.5 | 30.8 | 83.1 | **36.3** |
| | - Reward Model | 78.0 | 31.6 | 84.3 | 34.8 |
| | - One-shot ICL | **79.2** | 30.2 | 83.3 | 36.0 |

**Tool Manipulation.** The results are shown in Table 3. We can see that our JiuZhang3.0-7B and 8B models outperform all the baseline methods by a large margin, indicating the effectiveness of our approach in this setting. The reason is that we synthesize massive math problems in this format, which can teach JiuZhang3.0 models to accurately utilize tools by generating programs. Besides, the mixed synthetic math problem from the natural language reasoning setting can also benefit the required capabilities for this setting. Different from the natural language reasoning setting, JiuZhang3.0-8B (based on LLaMA-3-8B) performs better than the 7B version (based on Mistral-7B). The reason may be that LLaMA-3-8B owns stronger code synthesis and tool manipulation capability than Mistral-7B.

### 4.3 Further Analysis

**Performance w.r.t. Pre-training Data Amount.** In this part, we study how the scaling of synthetic data amount affects the model performance. We train Mistral-7B and LLaMA-3-8B using varying ratios of our synthetic entire dataset, *i.e.,* 20%, 40%, 60%, 80%, 100%, and report the performance on GSM8k, MATH, and ASDiv under the natural language reasoning setting. For comparison, we also show the results of the best-performed base LLM, *i.e.,* DeepSeekMath-Base-7B.

As shown in Figure 3, with the increasing of the training data ratio, the performance of our model improves consistently. Based on Mistral-7B, it can outperform the best-performed baseline using only 80% or 60% of the pre-training data, indicating the high quality of our synthetic pre-training data. Based on LLaMA-3-8B, it can perform better than the baseline using 40% or even 20% data, and the performance is consistently better than using Mistral-7B. It demonstrates that LLaMA-3-8B can better adapt into our synthetic data. Besides, the performance of our model can surpass the baseline more on MATH, which is a very complex dataset consisting of competitive problems, exhibiting the superiority of our method for improving the advanced mathematical reasoning capability.

**Ablation Study.** We conduct the ablation study to verify the effectiveness of key components in our proposed method. We test the following variations based on our approach, *i.e.,* (1) *w/o Prompt Set*: uses a simple prompt for guiding data synthesis instead of our crafted prompt set; (2) *w/o Math-related Texts*: directly synthesizes the math problems without math related texts; (3) *w/o Boosting Retraining*: uses the data synthesis model without retraining; (4) *w/o Value Estimation*: ignores the estimated value but randomly samples the instances for boosting training; (5) *w/o using GPT-4 for Boosting*: directly uses the high-value instance for boosting data synthesis model instead of using GPT-4. Limited by the computing resource, we conduct the ablation study under the natural language reasoning setting, and use 100k synthetic instances and randomly select 50k instructions from the instruction set. We report the results on GSM8K, MATH, ASDiv and CARP-en.

As shown in Table 4, all the variations mostly underperform the original model, indicating the effectiveness of all the components. Besides, the variation w/o using GPT-4 for Boosting performs slightly better in GSM8k and CARP, but degrades a lot in MATH (32.8⟶27.8). A possible reason is that it can benefit from the selected high-value data. But without the help of GPT-4, it can not synthesize helpful complex math problems for the competitive problems within MATH dataset.

**Variation Study for Data Synthesis LLMs.** To verify the effectiveness of our trained data synthesis LLM, we conduct the variation study using other existing LLMs for synthesizing the pre-training data. We select the following four LLMs, *i.e.,* ChatGPT, Mixtral-8×7B, DeepSeekMath-RL-7B, and LLaMA-3-8B-Instruct to replace our data synthesis LLM. We follow the efficient test setting in the ablation study, and report the results on GSM8K, MATH, ASDiv and CARP-en.

As shown in Table 4, all the variations mostly perform worse than the original model. It demonstrates that existing LLMs without adapted training might not be suitable to directly synthesize the data for pre-training. Besides, the performance of all the variation degrades a lot in MATH, which consists of complex competitive problems. It indicates that these existing LLMs are hard to synthesize the data that is useful for improving the performance in solving complex math problems.

**Variation Study for Data Selection Strategies.** To study the effectiveness of our gradient-based data selection strategy, we implement the following variations that replace it by other methods, *i.e.,* (1) *Random Sampling*: randomly samples the same number of instances; (2) *Perplexity*: selects the instances with lowest perplexity evaluated by Mistral-7B; (3) *Reward Model*: uses a well-trained reward model [36] for scoring; (4) *One-shot ICL*: concatenates the synthetic math problem and solution with the downstream task data to construct the one-shot in-context learning (ICL) example, and computes the decrease of loss as the estimated value [62]. We follow the efficient test setting.

As shown in Table 4, our original model mostly performs the best among all the variations, indicating the superiority of our gradient-based strategy. Whereas, the variation using one-hot ICL performs relatively better than others, and achieves the best performance on GSM8k. As the problems in GSM8k typically require more natural language reasoning steps, the ICL loss can well detect the instances with helpful context for solving these problems. However, it performs not well on MATH, where the math problems are complex and require using more math symbols and formulas.

## 5  Conclusion

In this paper, we proposed an efficient way to improve the mathematical reasoning of LLMs, where we trained a small LLM to synthesize sufficient high-quality math problems for pre-training. Concretely, we crafted a set of prompts that cover the knowledge and difficulty levels of human education stages, and selected the high-value math-related texts for downstream math-related tasks via the gradient-based strategy. Then, we fed them into GPT-4 to create the knowledge distillation dataset, which can better teach the data synthesis model to generate diverse and useful math problems. We utilized the synthetic data to pre-train JiuZhang3.0, and the whole process only required to invoke GPT-4 API 9.3k times and pre-train on 4.6B data. JiuZhang3.0 achieved state-of-the-art performance on several datasets under the natural language reasoning and tool manipulation settings, surpassing competitive LLMs that requires much larger cost on data synthesis or pre-training.

## Acknowledgments and Disclosure of Funding

This work was partially supported by Beijing Natural Science Foundation under Grant No. L233008 and 4222027, National Natural Science Foundation of China under Grant No. 62222215.

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

Table 5: The price of the potential service during the training procedure.

| Items | Details |
|---|---|
| **OpenAI API** | 30.00 USD per 1M tokens for input, and 60.00 USD per 1M tokens for output. |
| **AWS GPU Server** | 40.96 USD per 8 × A100 per hour. |

Table 6: The estimated cost of different LLMs using the official GPT-4 API and 8 nodes of 8×A100 GPU servers for training.

| Models | #API Calling (Token) | | | #Server Time (Hour) | | Total Expenses |
|---|---|---|---|---|---|---|
| | Input | Output | Data Num. | Synthesizing | Training | |
| **KPMath-DSMath-7B** | 1090 | 218 | 865K | - | - | 39,599 USD |
| **DeepSeekMath-7B-RL** | - | - | - | - | 160 | 52,428 USD |
| **JiuZhang3.0-7B** | 300 | 877 | 10K | 14 | 10 | 8,480 USD |

# A  Cost Estimation

To estimate the expenses of previous work and our proposed JiuZhang3.0, we survey the price of potential service during the entire procedure, including calling the OpenAI API for GPT-4 and renting the AWS GPU server for LLMs training. The details of the price are presented in Table 5. For a fair comparison, we assume that GPT-4 is utilized to synthesize training data, and 8 nodes of 8 × A100 GPU servers (64 GPUs in total) are leveraged for LLMs training.

For our data synthesis process, the average length of the prompting (including selected math-related texts) is about 300 tokens, and the average length of problems and solutions is 877 tokens. Under the setting of 64 GPUs, we spend 4 hours selecting valuable data for natural language reasoning and tool manipulation in total, and 10 hours synthesizing the 4.6B pre-training corpus. We train the JiuZhang3.0-7B model for 10 hours, including both pre-training and fine-tuning.

As the dataset and the construction process of KPMath are not publicly available, we adopt the average length of synthesis prompts, problems, and solutions in MetaMathQA for estimating the input and the output tokens, respectively. We estimate the training time on 120B tokens for a 7B model as 160 hours. Since the training details in the RLHF stage of DeepSeekMath-7B-RL are not publicly available, we do not count the fine-tuning cost for the KPMath model and the DeepSeekMath model. In this case, we estimate the expenses of the LLMs training process as follows,

$$\text{API Expenses} = (\text{Avg. Input Length} \times \text{API Input Price}$$
$$+ \text{Avg. Output Length} \times \text{API Output Price}) \times \text{Num Data}, \tag{6}$$

$$\text{Server Expenses} = \text{Num Nodes} \times \text{Price per Node} \times \text{Training Time}, \tag{7}$$

$$\text{Total Expenses} = \text{API Expenses} + \text{Server Expenses}. \tag{8}$$

The details and estimation of the expenses for different LLMs are present in Table 6.

# B  Fine-tuning Data

After pre-training, we collect a set of open-source math-related instructions to fine-tune JiuZhang3.0. For natural language reasoning, we collect the training sets of MATH [63], GSM8k [34], CARP [33], and open-source synthetic datasets based on them, *i.e.,* MetaMATH [10], MMIQC [41], Math-Shepherd (without problems from original MATH test sets) [64], Orca-MATH [11]. Besides, we also collect the positive examples from the PRM800k dataset (without problems from original MATH test sets) [35], and the TAL-SCQ5K dataset consisting of multi-choice questions [3]. Whereas, we observe that the varying data styles in the above datasets might cause the LLM outputs to be irregular. Thus, we utilize a unified prompt DeepSeekMath-7B-RL, to synthesize 700k solutions for the problems from the above datasets.

---

[3] https://github.com/math-eval/TAL-SCQ5K

For tool manipulation, we use the synthetic datasets, *i.e.,* OpenMathInstruct-1 [65] and MMOS [66], consisting of a mixture of text reasoning and code blocks executed by a Python interpreter.

## C    Evaluation Datasets

We test our JiuZhang3.0 and baseline methods in the following two settings for evaluating the mathematical reasoning capability.

• *Natural Language Reasoning*: we prompt LLMs to perform multi-step reasoning via natural language, and select the following publicly avaibable datasets: GSM8k [34] contains grade school math problems to test the basic arithmetic and reasoning ability. MATH [63] and CARP-en [33] consist of complex competition-level problems, and CARP-en is the English version of the original CARP dataset. ASDiv [67], MAWPS [68] and SVAMP [69] are grade-school math word problem (MWP) datasets, and SVAMP focuses on the robust reasoning ability. Besides, we also consider the following datasets with different data formats or related to other interdisciplinary field, *i.e.,* TabMWP, AQuA, SAT-Math, MathQA, MMLU-STEM. AQuA [70], SAT-Math [71], MMLU-STEM [72] are composed of multiple-choice questions for human exams across math and other STEM disciplines. OCW-Math [73] is a challenging dataset containing undergraduate-level math and science problems.

• *Tool Manipulation*: we prompt LLMs to manipulate external tools via Python to solve the problems, and select GSM8k, MATH, GSM-Hard, SVAMP, TabMWP, ASDiv, and MAWPS for testing. GSM-Hard [27] replaces the numbers in the questions of GSM8K with larger numbers to increase the difficulty of calculation. TabMWP [74] is an open-domain MWP dataset containing tabular data.

## D    Baseline Methods

We consider diverse types of baseline methods for comparison.

• *Closed-source LLMs*: ChatGPT and GPT-4 [2];

• *Larger LLMs (>20B)*: Qwen-1.5-110B [75], Qwen-1.5-72B [75], Deepseek-LM-67B [76], Mixtral-8×7B [19], Llemma-34B [7], Intern-Math-20B [9], MAmmoTH2-8×7B-Plus [13], ChatGLM-Math-32B [77];

• *Smaller LLMs (<10B)*: DeepSeek-7B [76], Qwen-1.5-7B [75], Mistral-7B [61], LLaMA-3-8B [18], Gemma-7B [78], and CodeLLama [79];

• *LLMs pre-trained on Math Corpus (<10B)*: Llemma-7B [7], InternLM-Math-7B [9], Rho-1-Math-7B [80], DeepSeekMath-7B [5];

• *LLMs fine-tuned on Math Instructions (<10B)*: MetaMath-Mistral-7B [10], WizardMath-7B-1.1 [49], Abel-7B-002 [81], Mistral-7B-MMIQC [41], Math-Shepherd-Mistral-7B-RL [64], DeepSeekMath-7B-Instruct [5], DeepSeekMath-7B-RL [5], Llama-3-8B-Instruct [18], MAmmoTH2 [13], KPMath-DSMath-7B [6]

• *LLMs fine-tuned on tool-augmented math instructions (<10B)*: MAmmoTH-7B-Mistral [39], MathCoder-7B-CL [82], ToRA-7B-Code [12], MARIO-OVM-7B [83], MMOS-CODE-7B [66], OpenMath-Mistral-7B [65], Rho-1-Math-7B-Code [80].

Our evaluation framework and in-context examples follow the existing work [6, 12, 84]. For general and math domain base models, we adopt the few-shot prompting method. For fine-tuned models, we adopt the zero-shot prompting method for open-ended natural language reasoning and tool manipulation tasks, and the few-shot prompting method for multiple choice problems. We cite the performance results reported in existing work [6, 12, 39, 65, 66, 77, 80, 82, 83].

## E    Implementation Details

**Data Synthesis Models.**    We train two data synthesis models for the natural language reasoning and tool manipulation settings, respectively. We first initialize them by training DeepSeekMath-7B-RL with 4k and 1.3k KD datasets, respectively. Then, we utilize the synthesis models to generate 100k problem-solution pairs for each one. During value estimation, we adopt the training set of GSM8k and MATH for natural language reasoning, and the 5k subset from a mixture of OpenMathInstruct

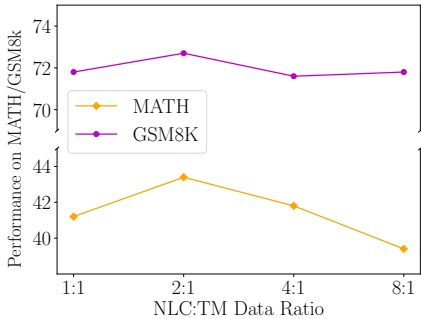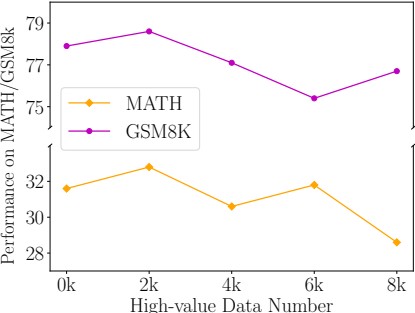

Figure 4: Hyper-parameter tuning results of the pre-training data proportion and high-value data amount, under the tool manipulation and natural language reasoning settings, respectively.

and MMOS for tool manipulation, as the instances from downstream math-related tasks. We select 2k the most valuable math texts in each setting for GPT-4 to boost the quality, and then add them into the KD dataset. During training, following existing work [14], we adopt a cosine learning rate schedule with a 0% warm-up ratio and select a learning rate of 1e-5 for 5 epochs and 10 epochs for natural language reasoning and tool manipulation, respectively.

**JiuZhang3.0.** Before training, we first filter out the instance from the synthetic dataset that have 10-grams overlap to the test set data, and also deduplicate data and the synthetic tool manipulation data containing unexecutable code. Then, we follow existing work that trains 7B, 8B, and 8×7B versions based on Mistral-7B, LLaMA-3-8B, and Mixtral-8×7B [13]. During training, we first pre-train it on our synthetic 4.6B math problem-solution pairs and then fine-tune it on the multi-source instruction set. We reuse the optimizer to initialize the fine-tuning stage and adopt the Warmup-Stable-Decay learning rate scheduler [85] with 3% warm-up ratio and 85% stable training ratio for 1 epoch in the whole training process. We set the maximum learning rate to 1e-5 and the minimum learning rate to 1e-6 with a total batch size of 512. To boost the training efficiency, we pack multiple instances in the same context window of the model and modify the attention to avoid mutual interference among difference instances. The maximum length of model is set to 2048. We train all models with BFloat16 numerical format, Flash Attention 2.0, DeepSpeed Stage 2 for 7B and 8B models, and Stage 3 for 8×7B models.

# F   Hyper-parameter Tuning

In this part, we conduct the experiments about tuning two important hyper-parameters, *i.e.,* pre-training data proportion, and the number of high-value data.

**Pre-train Data Proportion.** In the synthetic pre-training data proportion, we should determine the ratio between the data from the natural language reasoning and tool manipulation settings (NLC : TM). We set it to 1:1, 2:1, 4:1, and 8:1, and control the total data amount unchanged (200k) for comparison. We follow the efficient test setting in Section 4.3, and report the MATH and GSM8k results under the tool manipulation setting, as there is only slight performance changes on natural language reasoning setting. As shown in Figure 4, 2:1 is more suited and leads to better performance than other proportions. The reason may be that smaller or larger proportion for tool manipulation data would cause underfitting or overfitting on the tool manipulation data, affecting the corresponding capability.

**Number of Selected High-Value Data.** We also study the effect of changing the number of selected high-value data according to the ranking, as more data requires larger cost for invoking the GPT-4 API. We set it to 2k, 4k, 6k, and 8k, and do not change other settings for fair comparison. We follow the efficient test setting in Section 4.3. As shown in Figure 4, using the top 2k high-value data achieves the best performance, and more data even causes the performance degradation. It indicates

that the data quality is more important than data amount for boosting the data synthesis capability, as using more data would inevitably involve low-quality data.

# G   Limitation

First, although we train a strong yet small LLM that can synthesize high-quality math problems for training, its capabilities on synthesizing the data for other domains or tasks might be relatively weaker. The reason is that we only use the math-related data to train it. In future work, we will try to train a general-purpose model, to enable the data synthesis for other requirements. Second, in this work, we only focus on mathematical reasoning capability, and our trained JiuZhang3.0 is also mainly for solving math problems. Limited by the computation resource, we do not test its performance on other complex reasoning tasks, *e.g.,* planning, commonsense reasoning. We will also conduct the corresponding experiments in the future. Third, the 4.6B pre-training data is still a large scale for training, and we do not perform data filter to control its quality. Future work should focus on reducing its scale by proposing better data filter strategy. Forth, we only utilize GPT-4 for knowledge distillation, but do not use other well-performed LLMs, *e.g.,* Claude 3, GLM-4, and the latest GPT-4o. More experiments should be conducted on these LLMs to study the effect of the teacher LLMs. Fifth, for cost estimation, our reported results are all estimated according to our experience, without the re-implementation of these methods. We also appeal to report the true cost of training LLMs in existing work.

# H Case Study

**Type:** Natural Language Reasoning – Grade School
**Math-Related Texts:**
Question
The Huka Falls on the Waikato River is one of New Zealand's most visited natural tourist attractions (see Figure 12.30). On average the river has a flow rate of about 300,000 Ls. At the gorge, the river narrows to 20 m wide and averages 20 m deep. (a) What is the average speed of the river in the gorge? (b) What is the average speed of the water in the river downstream of the falls when it widens to 60 m and its depth increases to an average of 40 m?
Question Image
1. 0.75 ms
2. 0.13 ms
Solution Video# OpenStax College Physics Solution, Chapter 12, Problem 5 (Problems & Exercises) (1:14)
View sample solution## Calculator Screenshots
Video Transcript
This is College Physics Answers with Shaun Dychko. At the gorge where the Waikato River of New Zealand narrows to a width of 20 meters and a height of 20 meters, we'2019re going to figure out what its speed is. And so the volume flow rate is equal to the cross sectional area of the river multiply by its
**Synthesis QA:**
Question: The Waikato River flows through a gorge where it narrows down to a width of 20 meters and a depth of 20 meters. The average flow rate of the river is 300,000 liters per second.
**Question:** What is the average speed of the river in this gorge?
`Answer:`
To find the average speed of the river in the gorge, we need to use the formula for the volume flow rate:

$$\text{Volume Flow Rate} = \text{Cross-Sectional Area} \times \text{Speed}$$

First, let'2019s calculate the cross-sectional area of the river:
Cross-Sectional Area = Width × Depth
Given:
- Width = 20 meters
- Depth = 20 meters
So,

$$\text{Cross-Sectional Area} = 20 \text{ meters} \times 20 \text{ meters} = 400 \text{ square meters}$$

Now, we know the volume flow rate and the cross-sectional area. The volume flow rate ($Q$) is 300,000 liters per second, which we need to convert into cubic meters because the area is in square meters:

$$1 \text{ liter} = 1 \text{ cubic decimeter} = 0.001 \text{ cubic meters}$$

So,

$$Q = 300,000 \text{ liters per second} = 300,000 \times 0.001 \text{ cubic meters per second} = 300 \text{ cubic meters per second}$$

We can now use the formula for the volume flow rate to find the speed ($v$):

$$Q = \text{Cross-Sectional Area} \times v$$

Rearranging to solve for $v$:

$$v = \frac{Q}{\text{Cross-Sectional Area}}$$

Substituting the values we have:

$$v = \frac{300 \text{ cubic meters per second}}{400 \text{ square meters}} = 0.75 \text{ meters per second}$$

So, the average speed of the river in the gorge is **0.75 meters per second**

## Good Case 2

**Type:** Natural Language Reasoning – Middle
**Math-Related Texts:**
Successive parabolic interpolation
Successive parabolic interpolation is a technique for finding the extremum (minimum or maximum) of a continuous unimodal function by successively fitting parabolas (polynomials of degree two) to a function of one variable at three unique points or, in general, a function of n variables at 1+n(n+3)2 points, and at each iteration replacing the öldestp̈oint with the extremum of the fitted parabola.
Advantages
Only function values are used, and when this method converges to an extremum, it does so with an order of convergence of approximately 1.325. The superlinear rate of convergence is superior to that of other methods with only linear convergence (such as line search). Moreover, not requiring the computation or approximation of function derivatives makes successive parabolic interpolation a popular alternative to other methods that do require them (such as gradient descent and Newton's method).
Disadvantages
On the other hand, convergence (even

**Synthesis QA:**

`Question:` Alex and Jamie are working on a science project and need to find the maximum height of a rocket launched upward. They record the height of the rocket at three different times:
- At $t = 0$ seconds, the height is 0 meters.
- At $t = 5$ seconds, the height is 25 meters.
- At $t = 10$ seconds, the height is 40 meters.
To find the maximum height, Alex and Jamie decide to use successive parabolic interpolation. They start by fitting a parabola to these three points and then finding the maximum point of this parabola.
1. Write the general form of a quadratic (parabolic) equation: $y = ax^2 + bx + c$. 2. Use the three points to create a system of equations to find the coefficients $a$, $b$, and $c$. 3. Find the vertex of the parabola to determine the maximum height.

`Answer:`
1. **General Form of a Quadratic Equation:**
The general form of a quadratic equation is:
$$y = ax^2 + bx + c$$

2. **Creating a System of Equations:**
We have three points:
$$(0, 0), \quad (5, 25), \quad (10, 40)$$
Plugging these points into the equation $y = ax^2 + bx + c$:
- For $(0, 0)$:
$$0 = a(0)^2 + b(0) + c \implies c = 0$$
- For $(5, 25)$:
$$25 = a(5)^2 + b(5) + 0 \implies 25 = 25a + 5b \quad \text{(Equation 1)}$$
- For $(10, 40)$:
$$40 = a(10)^2 + b(10) + 0 \implies 40 = 100a + 10b \quad \text{(Equation 2)}$$

3. **Solving the System of Equations:**
Use Equation 1:
$$25 = 25a + 5b \quad \text{(Equation 1)}$$
Simplify:
$$5 = 5a + b \quad \text{(Equation 3)}$$
Use Equation 2:
$$40 = 100a + 10b \quad \text{(Equation 2)}$$
Simplify:
$$4 = 10a + b \quad \text{(Equation 4)}$$
Subtract Equation 4 from Equation 3:
$$(5a + b) - (10a + b) = 5 - 4 \implies -5a = 1 \implies a = -\frac{1}{5}$$
Plug $a = -\frac{1}{5}$ back into Equation 3:
$$5 = 5\left(-\frac{1}{5}\right) + b \implies 5 = -1 + b \implies b = 6$$

4. **Vertex of the Parabola:**
The vertex of a parabola $y = ax^2 + bx + c$ is given by:

$$x = -\frac{b}{2a}$$

Here, $a = -\frac{1}{5}$, $b = 6$, and $c = 0$:

$$x = -\frac{6}{2\left(-\frac{1}{5}\right)} = -\frac{6}{-\frac{2}{5}} = 15$$

So, the maximum height occurs at $t = 15$ seconds. However, this $t$ value exceeds the range of our data points. In practical interpolation methods, this would indicate a correction point is needed beyond the given data. But for demonstration, let's assume $t = 15$ is within bounds (extrapolation for students):
Plug $t = 15$ back into the original equation with adjusted coefficients:

$$y = -\frac{1}{5}(15)^2 + 6(15) = -\frac{1}{5}(225) + 90 = -45 + 90 = 45$$

Thus, the maximum height is **45 meters**

**Type:** Natural Language Reasoning – High
**Math-Related Texts:**
Let $k$ be a positive integer. Find all polynomials with real coefficients which satisfy the equation $P(P(x)) = (P(x))^k$. Let $k$ be a positive integer. Find all polynomials with real coefficients which satisfy the equation

$$P(P(x)) = (P(x))^k .$$

I simply don't even know how to think about this problem. I've tried simple stuff just to get my head on the problem. For example for $P(x) = x^n$ I have $P(P(x)) = (P(x))^n$, and I think that any polynomial $P(x) = x^n + x^{n-1} + \cdots + c$ can't be a solution as I would have $P(x) = P(x)q_1(x) + R$. After that I simply stare at the problem. Can you guys give some help ? Note: I would like to understand how to tackle these kind of problems, so I would be really grateful if you would explain the thinking process behind the solution. (This is optional, so feel free to give an answer as you prefer.) Thanks in advance.
First we note that only the constant polynomials $P \equiv 0$ and $P \equiv 1$, as well als $P \equiv -1$ for odd $k$, satisfy the equation. (This holds true for $k > 1$; for $k = 1$ any constant polynomial will do.) Now if $P$ is not constant, then the range $Y = \{P(x) : x \in \mathbb{R}\}$ of $P$ is an infinite set. For every $y \in Y$ we have $P(y) = y^k$, which means that the polynomial $Q(x) = P(x) - x^k$ is zero on $Y$. Since $Y$ is infinite, this implies that $Q$ is the zero polynomial. In conclusion, $P(x) = x^k$.
**Synthesis QA:**
`Question:` Consider the polynomial $P(x) = x^2$. We want to find out how $P$ behaves when we apply it twice, $P(P(x))$, and compare it with $P(x)$ raised to some power, $(P(x))^k$.
Let's explore the following scenario:
Given $P(x) = x^2$, find all values of $k$ such that the equation $P(P(x)) = (P(x))^k$ holds for all real numbers $x$. `Answer:`
First, we evaluate $P(P(x))$:

$$P(P(x)) = P(x^2) = (x^2)^2 = x^4$$

Next, we need to express $P(x)$ raised to some power $k$:

$$(P(x))^k = (x^2)^k = x^{2k}$$

We want $P(P(x)) = (P(x))^k$:

$$x^4 = x^{2k}$$

This equation holds for all real numbers $x$ if and only if the exponents are equal. Therefore, we set up the following equation:

$$4 = 2k$$

Solving for $k$:

$$k = \frac{4}{2} = 2$$

Thus, the equation $P(P(x)) = (P(x))^k$ holds for $k = 2$.
Now let's verify this:
For $k = 2$:

$$P(P(x)) = x^4$$
$$(P(x))^k = (x^2)^2 = x^4$$

Since both sides of the equation are equal:

$$x^4 = x^4$$

This equation holds true for all real numbers $x$.
Therefore, the only value of $k$ that makes $P(P(x)) = (P(x))^k$ true for all real numbers $x$ is $k = 2$.

## Good Case 4

**Type:** Tool Manipulation – Grade School
**Math-Related Texts:**
Refer to a unit in the Units package by indexing the name or symbol with the context, for example, bit[information] or bps[information]; or, if the context is indicated as the default, by using only the unit name or symbol, for example, bit or bps. The units of information are defined as follows. A crumb is defined as 2 bits. A nibble is defined as 4 bits.

**Synthesis QA:**
Question: In a digital communication system, information is transmitted in units called bits. A crumb is defined as 2 bits, and a nibble is defined as 4 bits. If a user sends a sequence of bits, and the sequence can be broken down into whole crumbs and nibbles, how many different ways can the user send exactly 18 bits?
Answer:

```
def count_ways_to_send_bits(total_bits):
    ways = 0
    for crumb in range(total_bits // 2 + 1):
        for nibble in range(total_bits // 4 + 1):
            if crumb * 2 + nibble * 4 == total_bits:
                ways += 1
    return ways
total_bits = 18
ways_to_send = count_ways_to_send_bits(total_bits)
print(ways_to_send)
```

```output
5
```

There are ⬚5 different ways the user can send exactly 18 bits.

## Good Case 5

**Type:** Tool Manipulation – Secondary School Competition

**Math-Related Texts:**

The generalization of this is 0: $I = -log(p(x))$ Mean and variance of continuous random variables

The mean is the sum of x * density probability function at that point. $\mu = \int_{-\infty}^{\infty} xp(x)$

The variance is the sum of distance to the mean times the density probability function at that point. $\sigma^2 = \int_{-\infty}^{\infty} (x - \mu)^2 p(x)$

Esperance of a probability distribution is the value we could expect to have if we repeat some experiment a lot of times. It's the sum of the value of the events ponderated by the probability of the event to occur.

**Synthesis QA:**

Question: Consider a continuous random variable $X$ with probability density function (pdf) given by: $p(x) = \begin{cases} \frac{1}{2}e^{-\frac{x}{2}} & \text{if } x \geq 0 \\ 0 & \text{if } x < 0 \end{cases}$  1. Calculate the mean ($\mu$) of the distribution.

2. Calculate the variance ($\sigma^2$) of the distribution.

Answer:

```python
import sympy as sp

# Define the variable and the pdf
x = sp.Symbol('x', real=True, positive=True)
pdf = (1/2) * sp.exp(-x/2)

# Calculate the mean
mean = sp.integrate(x * pdf, (x, 0, sp.oo))

# Calculate the second moment about the origin
second_moment = sp.integrate(x**2 * pdf, (x, 0, sp.oo))

# Calculate the variance
variance = second_moment - mean**2

mean_value = mean.evalf()
variance_value = variance.evalf()

mean_value, variance_value
```

```output
(2.0, 4.0)
```

The mean ($\mu$) of the distribution is $\boxed{2.0}$, and the variance ($\sigma^2$) of the distribution is $\boxed{4.0}$.

## Bad Case 1

**Type:** Natural Language Reasoning – AMC12

**Math-Related Texts:**

Julia Robinson Mathematics Festival

The Julia Robinson Mathematics Festival (JRMF) is an educational organization that sponsors locally organized mathematics festivals and online webinars targeting K–12 students. The events are designed to introduce students to mathematics in a collaborative and non-competitive forum.[1][2][3]

AbbreviationJRMF FounderNancy Blachman Location • c/o SEE, 23564 Calabasas Road, Suite 201, Calabasas, CA 91302 Coordinates37°22'48.57"N 121°54'25.98"W Key people Nancy Blachman, Founder Daniel Kline, Executive Director Website-jrmf.org

History

In the 1970s, Saint Mary's College of California produced a mathematics contest that was popular with secondary schools throughout the San Francisco Bay Area.[4] In 2005, Nancy Blachman attended an education forum sponsored by the Mathematical Sciences Research Institute (MSRI) and remembered how the Saint Mary's contest had inspired her as a student.[5] Unfortunately, the contest no longer existed. Seeking to possibl

**Synthesis QA:**

**Question:** At the Julia Robinson Mathematics Festival, students are challenged with a geometry puzzle. The problem is as follows: A square $ABCD$ with side length 10 is inscribed in a circle. Points $P$ and $Q$ lie on the circle such that $P$ is on the same side of line $AD$ as $C$, and $Q$ is on the same side of line $AB$ as $C$. If $AP = 16$ and $AQ = 12$, find the length of $PQ$.

**Answer:** To solve this problem, we'll start by finding the radius of the circle, using the fact that the square is inscribed in the circle. Then, we'll use properties of circles and the given lengths to determine the length of $PQ$.

1. **Finding the radius of the circle:**

Since the square $ABCD$ is inscribed in the circle, the diameter of the circle is equal to the diagonal of the square. The diagonal of a square with side length $s$ can be calculated using the Pythagorean theorem:

$$\text{Diagonal} = s\sqrt{2}$$

For the given square with side length 10, the diagonal is:

$$\text{Diagonal} = 10\sqrt{2}$$

Thus, the radius of the circle, which is half the diagonal, is:

$$\text{Radius} = \frac{10\sqrt{2}}{2} = 5\sqrt{2}$$

2. **Using the Power of a Point Theorem:**

The Power of a Point Theorem states that for a point $P$ and a point $Q$ in a circle with center $O$ and radius $r$, if $P$ and $Q$ are on the same side of a chord $CD$ passing through $O$, then the product of the distances from $P$ and $Q$ to $CD$ is equal to the square of the distance from $O$ to $CD$.

In our case, $CD$ is the diameter of the circle, and $O$ is the center of the circle. Point $P$ is 16 units from $A$ on side $AD$, and $Q$ is 12 units from $A$ on side $AB$. $AP = 16$ and $AQ = 12$. Since $P$ and $Q$ are on the circle and the diameter passes through the center $O$, the Power of a Point Theorem gives:

$$AP \cdot AQ = OP^2$$

$$16 \times 12 = OP^2$$

$$OP = \sqrt{192} = 8\sqrt{3}$$

3. **Determining the length of $PQ$:**

Finally, we use the radius of the circle and the calculated power of point distance to find $OP$ to determine $PQ$. Since $O$ is the midpoint of $PQ$ and $OP = OQ = 8\sqrt{3}$:

$$PQ = 2 \times OP = 2 \times 8\sqrt{3} = 16\sqrt{3}$$

Thus, the length of $PQ$ is $16\sqrt{3}$ units.

This problem challenges students to use geometric principles (properties of circles and squares) and algebraic reasoning to solve for lengths in a geometric configuration, enhancing analytic and problem-solving skills suitable for the AMC 12 level.

### Bad Case 2

**Type:** Tool Manipulation – Secondary School Competition
**Math-Related Texts:**
0Nick D. Jan 5 '17 at 14:48
• I guess it comes from a wifi code.Are you a chinese student? –0Renascence5. Jan 6 '17 at 8:50
HINT:
Observe that the function is odd and the interval $[-5, 5]$ is symmetric about $0$. See more in this question:
Definite integral of an odd function is 0 (symmetric interval)

$$x^4 + 2x^2 + 1 = \left(x^2 + 1\right)^2$$

so we have an integral of the form

$$\int \frac{x^3}{(x^2 + 1)^2} \sin^2 x\, dx$$

we can clearly see that

$$f(x) = \frac{x^3}{(x^2 + 1)^2} \sin^2 x \implies f(-x) = -f(x)$$

i.e.
```
```
Guidelines for each section:

**Synthesis QA:**
`Question:` Consider the function $f(x) = \frac{x^3}{(x^2+1)^2} \sin^2 x$ over the interval $[-5, 5]$. We are interested in evaluating the definite integral of this function with respect to $x$. Given the properties of the function $f(x)$, which is odd, and the symmetric interval about the origin, determine the value of the integral.
`Answer:`

```
import sympy as sp

# Define the variable
x = sp.symbols('x')

# Define the function
f = (x**3) / (x**2 + 1)**2 * sp.sin(x)**2

# Since the function is odd over a symmetric interval, the integral is zero
integral_result = sp.integrate(f, (x, -5, 5))

print(integral_result)
```
```output
0
```
The answer is $\boxed{0}$.

# I   Prompts

## I.1   Prompts for Natural Language Reasoning Data Synthesis

### Prompts for Grade School-Level Problem

**Instruction**
Create an age-appropriate math word problem for grade school students based on the provided math content.

**Math Content**
[Math Text Placeholder]

**Guidelines**
[Problem]: Craft a concise math word problem suitable for grade school, focusing on basic arithmetic operations (addition, subtraction, multiplication, division), number sense, simple shapes, or introductory measurements. Use relatable, real-world scenarios appropriate for the age group. Ensure the problem is purely text-based and solvable without images. [Solution]: Provide a clear, step-by-step solution to the problem using simple language that a grade school student could understand. Explain the reasoning behind each step.

## Prompts for Middle School-Level Problem

**Instruction**
Create an middle school level math problem and solution based on the provided math content excerpt.

**Math Content**
[Math Text Placeholder]

**Guidelines**
[Problem]: Create a self-contained problem for middle school student that directly incorporates a concept from the provided math content. Target a difficulty level appropriate for grades 6-8 (ages 11-14), assuming knowledge of arithmetic, pre-algebra, basic probability/statistics, and geometry. Ensure the problem is fully text-based and solvable without images. Use concepts typically covered by the end of 8th grade.
[Solution]: Provide a detailed, step-by-step solution that demonstrates the mathematical reasoning from problem statement to conclusion, around 250-350 words long. Utilize LaTeX formatting for all mathematical expressions. Explain each step to reinforce the underlying math principles being applied.

## Prompts for High-Level Problem

**Instruction**
Inspired by the provided math content extract, create high school-level math problem that combines concepts from at least two math subjects.

**Math Content**
[Math Text Placeholder]

**Guidelines**
[Problem]: Draft a self-contained math problem for high school students based on the given math content. The problem should require knowledge from one of these subjects: Algebra I and II, Pre-Calculus, Calculus, Geometry, Trigonometry, Statistics and Probability. Ensure the problem is fully text-based and solvable without images. Use concepts typically covered by the end of 11th grade.
[Solution]: Provide a detailed, step-by-step solution that demonstrates the mathematical reasoning from problem statement to conclusion, around 250-350 words long. Utilize LaTeX formatting for all mathematical expressions. Explain each step to reinforce the underlying math principles being applied.

## Prompts for High-Level Problem

**Instruction**
Inspired by the math content, create a college-level math problem.

**Math Content**
[Math Text Placeholder]

**Guidelines**
[Problem]: Draft a self-contained, college-level math problem inspired by the math content. It should be intellectually stimulating and designed for an audience familiar with advanced mathematics, such as Calculus, Linear Algebra, Abstract Algebra, etc. Ensure the problem includes all necessary information for solving it. Aim for a problem statement around 100-150 words. [Solution]: Provide a step-by-step solution to your problem, around 250-350 words long. The solution should clearly explain the reasoning, mathematical principles, and steps used. Call out any key theorems or properties being applied at each step.

## Prompts for AMC 8-Level Problem

**Instruction**
Inspired by the provided math content, craft a math problem suitable for the AMC 8 competition, engaging top grade students with a challenge in areas of arithmetic, algebra, counting, geometry, logic, number theory, or probability. Ensure the problem stimulates problem-solving techniques and heuristics within a text-based and solvable format, avoiding advanced subjects like calculus or physics.

**Math Content**
[Math Text Placeholder]

**Guidelines**
[Problem]: Design a compelling, self-contained math problem for AMC 8 contestants inspired by the math content, incorporating elements of basic math disciplines. The problem should be approachable through logical reasoning and fundamental problem-solving strategies. Ensure the problem is fully text-based, solvable without the aid of images, and has a difficulty level appropriate for the AMC 8 competition.
[Solution]: Provide a detailed, step-by-step solution that is educational and tailored to the AMC 8 audience, around 250-350 words long. Explain the logic and reasoning behind each step thoroughly, using clear and age-appropriate language and terminology to facilitate understanding. Highlight the use of problem-solving techniques and heuristics employed in the solution.

## Prompts for AMC 12-Level Problem

**Instruction**
Inspired by the provided math content, develop a math problem suitable for the AMC 12 competition, targeting high-school students with capabilities in advanced areas like algebra, geometry, trigonometry, counting, probability, and number theory. The problem should challenge students to utilize sophisticated problem-solving skills and mathematical reasoning. Focus on concepts from the given content, but adapt them into an original problem. Avoid delving into higher-level college mathematics topics beyond the AMC 12 scope.

**Math Content**
[Math Text Placeholder]

**Guidelines**
[Problem]: Formulate a compelling and complex math problem for AMC 12 participants inspired by the provided math content. The problem should encourage students to employ advanced logical reasoning and problem-solving strategies related to the given concepts. Craft it to be solvable in the AMC 12 format and difficulty level.
[Solution]: Present a comprehensive, step-by-step solution that both solves the problem and educates the student, around 250-350 words long. Clearly articulate the reasoning and methods used at each step, providing insight into the problem-solving process. Use language that challenges yet instructs high school students looking to improve their skills. Take care to format any equations properly using LaTeX or appropriate notation.

## Prompts for AIME-Level Problem

**Instruction**
Inspired by the math content, create a math problem appropriate for advanced high school mathematics competitions like the AIME. The problem should challenge students in core areas tested on the AIME such as algebra, geometry, combinatorics, number theory and probability. Encourage creative problem-solving and deep mathematical thinking using pre-calculus level techniques. Avoid delving into college-level topics like abstract/linear algebra, topology, multivariable calculus, or advanced physics. Any physics concepts used should be kept at a basic mechanics level.

**Math Content**
[Math Text Placeholder]

**Guidelines**
[Problem]: Design an insightful and challenging problem suitable for AIME participants inspired by the math content. Focus on core AIME topics like algebra, geometry, combinatorics, number theory and probability. The problem should be solvable using clever applications of high school math and mathematical reasoning, without requiring knowledge of advanced college-level mathematics or physics. Aim for a difficulty level on par with actual AIME questions. [Solution]: Provide a detailed, step-by-step solution that would be enlightening to an advanced high school student, around 250-350 words long. Explain the reasoning process and techniques applied in each step. Use language that is mathematically sophisticated yet understandable to a well-prepared AIME participant. Justify any key insights or creative leaps in the solution process.

## I.2   Prompts for Tool Manipulation Data Synthesis

## Prompts for Tool Manipulation where Grade School or Secondary School Competition Level is Determined by Example

**Instruction**
Please gain inspiration from the following random math content to create a high-quality math problem and solve it with Python code. Present your output in two distinct sections: [Problem Description] and [Solution].

**Math Content**
[Math Text Placeholder]

**Guidelines**
1. [Problem Description]: This should be **completely self-contained**, providing all the contextual information one needs to understand and solve the problem.
2. [Solution]: Offer a comprehensive, **correct** solution that accurately addresses the [Problem Description] you provided using Python code. Summarize in natural language in the end and put the answer in /boxed.
**Example**
[Problem Description]
[Example Question Placeholder]
[/Problem Description]
[Solution]
[Example Python Solution Placeholder]
[/Solution]

