# OpenReview forum: "JiuZhang3.0: Efficiently Improving Mathematical Reasoning by Training Small Data Synthesis Models"
_NeurIPS.cc/2024/Conference — NeurIPS 2024 poster_

### Official Review · Reviewer_rcgh · 2024-07-13

**Soundness:** 4
**Presentation:** 3
**Contribution:** 4
**Rating:** 8
**Confidence:** 4

**Summary:**

The paper proposes distilling the data synthesis ability from the strong (and large) model to the small model and using the small model to synthesize (continual) pre-training data, focusing on mathematical problem-solving with natural language reasoning and tool-integrated reasoning.

Specifically, the paper 1) elaborately designs a prompt set covering different human education stages/difficulty levels, prompts GPT-4 with the designed prompts and diverse math-related texts to synthsize mathematical QA pairs, 2) uses the data synthesis samples (math-related text + designed prompt + QA pair) from 1) to train the small data synthesis model from DeepSeekMath-7B, leverages it to generate large-scale candidates, select the most valuable samples using LESS and utilizes GPT-4 to re-generate the valuable QA pairs, 3) re-trains the data synthesis model with the data merged from 1) and 2) and uses it to synthesize large-scale QA pairs, 4) continually pre-trains various base models with the data from 3), 5) finally, fine-tunes the continually-pre-trained model with mixed open-source instruction datasets.

The paper

- proposes an effective and efficient pipeline to synthesize large-scale high-quality instruction data;
- empirically validates the effectiveness of scaling up instruction data for reasoning tasks;
- conducts extensive ablation studies, providing many insights like:
  - diversity of prompts for data synthesis, reference text, and strong/specialized data synthesis models are important;
  - re-training with self-generated data synthesis samples is actually harmful.

**Strengths:**

- The scaling effect of instruction data is an important empirical question. The paper is one of the first works to scale instruction data over 5M pairs, showing the feasibility and effectiveness of scaling up instruction data (for reasoning tasks).
- Instruction data synthesis usually depends on ground truth answer supervision signals, which is not always accessible. The paper validates the effectiveness of scaling synthetic instruction data without ground truth answers to a large scale.
- Direct distillation / data synthesis from strong but large models is effective but expensive. The paper reveals that the data synthesis abilities of strong but large models like GPT-4 can be effectively distilled into small models like DeepSeekMath-7B, making the data synthesis more efficient.
- JiuZhang3.0 achieves performance superior to or comparable with previous SotA on various mathematical reasoning datasets, under both natural language reasoning and tool manipulation settings. Especially,
  - JiuZhang3.0-7/8B are respectively based on Mistral-7B and LLaMA-3-8B, which are conventionally thought weaker than DeepSeekMath-7B, but achieve impressing performance slightly better than DeepSeekMath-7B-RL. (JiuZhang3.0 based on DeepSeekMath-7B should be expected to achieve even better performance.)
  - Scaling curves of performance linear to the training dataset size until a rather large scale (\~5M) are shown in Figure 3, which should be better than usual log-linear ones.
- Extensive experiments on various base models and benchmarks and comprehensive ablation studies are done to validate the effectiveness of the pipeline.
- The paper is generally well written to be clear and detailed.

**Weaknesses:**

- The paper achieves its best performances with a 2-stage instruction tuning on pre-trained models. Still, it doesn’t involve continual pre-training, which should be rather important for models’ reasoning abilities as proved by works like DeepSeekMath. It might need further consideration about **whether continual pre-training on large-scale synthetic instruction data before fine-tuning is compatible with or necessary to be added to existing training pipelines to achieve the best final performance (for reasoning tasks)**. Pre-training and RL should be out of this work’s scope. However, it would be better to further clarify the impacts of 1) continual pre-training on domain-specific corpora, 2) continual pre-training on large-scale instruction data, 3) final fine-tuning on additional instruction datasets, and their combinations.
  - According to the baseline results from DART-Math, directly fine-tuning on DeepSeekMath-7B-RL-synthesized data only can produce competitive performance (on GSM8K and MATH under the natural language reasoning setting). **Ablation studies of performance by 2) and 3)** only should be done to better validate the compatibility/necessity of the pipeline.
  - **To consider 1) continual pre-training on domain-specific corpora**, it is unrealistic to conduct by yourselves, but a possible workaround could be to make full use of the resources provided by DeepSeekMath: DeepSeekMath-7B is continually pre-trained from DeepSeek-Coder-Base-v1.5. By comparing performances on reasoning benchmarks of applying 2/3/2+3 on DeepSeek-Coder-Base-v1.5/DeepSeekMath-7B and the two models themselves, a more comprehensive study on the impacts of these training stages can be done.
- The abilities of the data synthesis model should be an important factor and the size of the knowledge distillation (KD) dataset could have a significant impact on the data synthesis abilities, but no related ablation studies have been done. It would be better to investigate **the scaling behavior of the KD dataset size**.
- Continual pre-training is usually done with LM loss without masking the prompt tokens (e.g. in DeepSeekMath). It is not mentioned **which type of loss is used for (continual) pre-training** in the paper and worth corresponding ablation studies.
- The upper limit of synthetic data is important. The student model could be weaker than the teacher model for data synthesis, so the results in the paper might not be the upper limit. It is so expensive and unrealistic for the authors to scale synthetic data to the same scale using GPT-4, but **more comparisons with dataset size controlled with previous works training on large-scale GPT-4-synthesized data (e.g. KPMath)** should be also helpful.

---

# After rebuttal and discussion

The authors resolve almost all the concerns.

I would improve my rating to 8 because JiuZhang3.0 has been validated to replace the continual pre-training stage in the standard SotA pipeline and shows great potential to combine further with continual pre-training like DeepSeekMath to effectively push forward the upper limit.

The main limitation is that the pipeline still relies on strong DeepSeekMath models for data synthesis, not fully eliminating the necessity of continual pre-training on large-scale corpora, which might be a bottleneck for improving the best models like GPT-4. So I could not give a rating of 9.

**Questions:**

Suggestions:

- The current setting is more like distillation from DeepSeekMath (we can substitute GPT-4 with human experts for the small KD dataset). The method could be more promising if it could help self-improvement/weak-to-strong generalization. I highly recommend adding experiments on training DeepSeekMath or stronger models in future versions.

Confusions:

- How is the reward model (Eurus-RM should be for reward to correctness) specifically used for data selection?
- How is the fine-tuning dataset synthesized with DeepSeekMath-7B-RL?
- How is data deduplication done for the (continual) pre-training data?

Typo:

- Line 734: remove deduplicate → deduplicate?

**Limitations:**

The limitations of this work are acceptable and the authors point out potential directions to address the limitations for future works.

---

> ### Author Rebuttal · Authors · 2024-08-04
>
> We sincerely thank the reviewer for the insightful suggestion and positive feedback, and list our responses for the weaknesses 1 below:
>
> **W1**. It might need further consideration about whether continual pre-training on large-scale synthetic instruction data before fine-tuning is compatible with or necessary to be added to existing training pipelines to achieve the best final performance (for reasoning tasks). Pre-training and RL should be out of this work’s scope. However, it would be better to further clarify the impacts of 1) continual pre-training on domain-specific corpora, 2) continual pre-training on large-scale instruction data, 3) final fine-tuning on additional instruction datasets, and their combinations.
> - **Reply to W1**. Recently, continual pre-training on large-scale instructions has been widely used, due to its effectiveness in improving domain/task-specific capability[1,2]. After it, by further fine-tuning on less high-quality instruction, LLMs would be able to understand and follow more complex domain/task-specific instructions. In this work, we name our synthetic data as math-specific pre-training corpus, as its scale (4.6B tokens) is much larger than commonly-used instruction datasets, and we also conduct SFT after training on it. In fact, we think this corpus should not be directly used for instruction-tuning, as we can not guarantee the accuracy of its contained data. Thus, it is necessary to add the fine-tuning process with reliable and accurate data after training on it. In future work, we will explore how to guarantee the accuracy of the data and compress our dataset into a relatively less instruction dataset specially for fine-tuning.
>
> [1] Yue, Xiang, et al. "Mammoth2: Scaling instructions from the web." arXiv preprint arXiv:2405.03548 (2024).
>
> [2] Cheng, Daixuan, et al. "Instruction Pre-Training: Language Models are Supervised Multitask Learners." arXiv preprint arXiv:2406.14491 (2024).
>
>
> **W1-a**. According to the baseline results from DART-Math, directly fine-tuning on DeepSeekMath-7B-RL-synthesized data only can produce competitive performance (on GSM8K and MATH under the natural language reasoning setting). Ablation studies of performance by 2) and 3) only should be done to better validate the compatibility/necessity of the pipeline.
> - **Reply to W1-a**. Following the suggestion of the reviewer, we add the ablation studies that train Mistral-7B only using the pre-trained or SFT data, respectively, and also report the performance of DeepSeekMath-Base and JiuZhang3.0-7B for comparison. As shown in the following Table, after using the pre-training data, our base model can outperform DeepSeekMath-Base, SOTA open-source backbone for mathematical reasoning, indicating the effectiveness of our pre-training data. Besides, by only using the SFT data, Mistral-7B+SFT underperforms JiuZhang3.0-7B in a large margin. It also demonstrates that our pre-training data can improve the potential of LLMs for learning the SFT data.
>
> |         | GSM8k | MATH |	SVAMP	| ASDiv | MAWPS | CARP |
> |------------|------------|------|---------|------------|------------|------|
> | DeepSeekMath-Base-7B	|64.1|	34.2|	73.7	|82.7	|92.7|	44.4|
> | JiuZhang3.0-7B-Base|	68.7	|40.2	|80.4	|86.5	|93.5	|40.2|
> | Mistral-7B + SFT	|87.5|	45.8|	87.9|	89.7	|97|	44.4|
> | JiuZhang3.0-7B |	88.6	|52.8|	90.4	|92.6	|97.3|	51|
>
> **W1-b & Q1**. To consider 1) continual pre-training on domain-specific corpora, it is unrealistic to conduct by yourselves, but a possible workaround could be to make full use of the resources provided by DeepSeekMath: DeepSeekMath-7B is continually pre-trained from DeepSeek-Coder-Base-v1.5. By comparing performances on reasoning benchmarks of applying 2/3/2+3 on DeepSeek-Coder-Base-v1.5/DeepSeekMath-7B and the two models themselves, a more comprehensive study on the impacts of these training stages can be done.
> - **Reply to W1-b & Q1**. Following the suggestion of the reviewer, we apply our synthetic pre-training data and collected SFT data, to train DeepSeek-Coder-Base-v1.5 and DeepSeekMath-7B-base. We report the results in the following Table, where we can see a consistent trend that Backbone+PT+SFT > Backbone+SFT > Backbone+PT > Backbone. It indicates that our overall framework works well on different backbones. The reason why Backbone+SFT > Backbone+PT is that the SFT data is typically more like or even derives from the downstream evaluation datasets. With the help of our synthetic pre-training data, the math reasoning capability can be enhanced more after SFT. Besides, we can see that our approach leads to more improvement on DeepSeekMath-7B-base than DeepSeek-Coder-Base-v1.5. It demonstrates that continually pre-training on more domain-specific data can boost the model capability more.
>
> |         | GSM8k | MATH |	SVAMP	| ASDiv | MAWPS | CARP |
> |------------|------------|------|---------|------------|------------|------|
> | DeepSeekMath-Base	|64.1|	34.2|	73.7	|82.7	|92.7|	44.4|
> | DeepSeekMath-Base + SFT|	87.1	|51.2|	86.5|	91.2|	96.6|	52.2|
> | DeepSeekMath-Base + PT	|77.6	|46.6	|87.4	|90	|94.8	|50.4|
> | DeepSeekMath-Base + PT SFT|	87.6|	52.6	|88.6|	93.1|	96.7|	52|
> |DeepSeekCoder-Base-v1.5|	41.3	|17.4	|68.5	|71	|85.2	|32.6|
> |DeepSeekCoder-Base-v1.5 + PT	|67.6|	43.2|	80.2|	86.5	|91.9|	42.3|
> |DeepSeekCoder-Base-v1.5 + SFT	|81.9	|40.8	|83.3	|88.3	|96.3|	43.9|
> |DeepSeekCoder-Base-v1.5 + PT SFT|	86.2|	47.4|	87.2|	90.9|	96.8|	49.6|
> | JiuZhang3.0-7B |	88.6	|52.8|	90.4	|92.6	|97.3|	51|
> |JiuZhang3.0-8B	| 88.6	|51	|89.4	|92.6|	97.1	|50.9|

---

> ### Author Response · Authors · 2024-08-04
> **Rebuttal by Authors (part-2)**
>
> We sincerely thank the reviewer for the insightful suggestion and positive feedback, and list our responses for the weaknesses 2-4 below:
>
> **W2**. The abilities of the data synthesis model should be an important factor and the size of the knowledge distillation (KD) dataset could have a significant impact on the data synthesis abilities, but no related ablation studies have been done. It would be better to investigate the scaling behavior of the KD dataset size.
> - **Reply to W2**. We report how the scaling the KD dataset size affects the model performance in Figure 4 from Appendix. Concretely, we set the number of selected high-value data into 2k, 4k, 6k, and 8k, and follow the test setting in ablation study using 100k and 50k samples for pre-training and SFT, respectively. As shown in Figure 4, using the top 2k high-value data achieves the best performance, and more data even causes the performance degradation. It indicates that the data quality is more important than data amount for boosting the data synthesis capability, as using more data would inevitably involve low-quality data. Besides, it also indicates that data synthesis is not a hard-to-learn capability for small LLMs. Therefore, we only use 10k data for training, which also reduces the cost of using GPT-4 APIs.
>
> **W3**. Continual pre-training is usually done with LM loss without masking the prompt tokens (e.g. in DeepSeekMath). It is not mentioned which type of loss is used for (continual) pre-training in the paper and worth corresponding ablation studies.
>
> - **Reply to W3**. We follow existing work[1,2] that adds a mask on the question tokens and only computes loss on the solution tokens during continual pre-training. Following the suggestion of the reviewer, we also test whether removing the mask would lead to better performance. Concretely, we follow the setting in the ablation study that conduct the experiments on Mistral-7B using 100k pre-training and 50k SFT samples. As shown in the following table, we can see that without the mask the performance degrades. The reason may be that the question tokens only depict the given condition of the math problems, which is not useful for improving the problem solving capability of LLMs.
>
> [1] Yue, Xiang, et al. "Mammoth2: Scaling instructions from the web." arXiv preprint arXiv:2405.03548 (2024).
>
> [2] Cheng, Daixuan, et al. "Instruction Pre-Training: Language Models are Supervised Multitask Learners." arXiv preprint arXiv:2406.14491 (2024).
>
> |         | GSM8k | MATH |	ASDiv | CARP |
> |------------|-----------|--------|------------|------------|
> | **Ours** |78.6|	32.8|	84.5|	36.2|
> | **Ours w/o Mask** | 78.5	|31.2|	79.7	|35.5|
>
> **W4**. The upper limit of synthetic data is important. The student model could be weaker than the teacher model for data synthesis, so the results in the paper might not be the upper limit. It is so expensive and unrealistic for the authors to scale synthetic data to the same scale using GPT-4, but more comparisons with dataset size controlled with previous works training on large-scale GPT-4-synthesized data (e.g. KPMath) should be also helpful.
> - **Reply to W4**. Following the suggestion of the reviewer, we control the number of training sample in KPMath and our approach being same, and compare the performance of using our approach and KPMath. We keep the scale of SFT data unchanged, but reduce the number of pre-training samples, until the same as KPMath. We report the results in the follow Table, where we can see that our approach can even perform better than KPMath. It indicates the quality of our synthetic data. The reason may be that our devised prompts and collected math-related text data help better guide the data synthesis process, enables it to outperform KPMath using GPT-4-synthesized data.
>
> |         | GSM8k | MATH |	SVAMP	| ASDiv | MAWPS |
> |------------|------------|------|---------|------------|------------|
> |**Ours - Same Data Size** |87.9	|49|	89.6	|91.3	|97.2|
> |**KPMath**|	83.9|	48.8|	81.5	|88.9|	94.8|

---

> ### Author Response · Authors · 2024-08-04
> **Rebuttal by Authors (part-3)**
>
> We sincerely thank the reviewer for the insightful suggestion and positive feedback, and list our responses for the questions below:
>
> **Q2**. How is the reward model (Eurus-RM should be for reward to correctness) specifically used for data selection?
> - **Reply to Q2**. The used reward model Eurus-RM can produce the reward score that reflects the human preference. For each question and its solution, we regard the question as the user input instruction, and utilize Eurus-RM to estimate whether the solution is helpful for it. In ablation study, we use the above way to replace our gradient-based value estimation strategy, to score all the synthetic candidates from the initialized synthesis model. Then, the top-ranked 2k samples are used to boost the model, and finally we use the boosted model to synthesize large-scale data for pre-training.
>
> **Q3**. How is the fine-tuning dataset synthesized with DeepSeekMath-7B-RL?
> - **Reply to Q3**. We directly feed the questions from all the collected instruction datasets into DeepSeekMath-7B-RL, and use a unified prompt to guide it. Fo each question, we first synthesize eight solutions. Then, if the dataset has provided the answer, we use rules to filter out the solutions with wrong answers. If not, we regard the mostly common answer from all the solutions as the correct one, and filter others. Finally, we keep at most 3 solutions per question, total 700k samples. The synthetic 700k samples help regulate the output format of LLMs, to avoid misleading by diverse formats from collected instruction datasets.
>
> **Q4**. How is data deduplication done for the (continual) pre-training data?
> - **Reply to Q4**. We use the Minhash algorithm on the question part for all the synthetic samples, and set the threshold into 0.7 for filtering the duplicate questions. Concretely, we use the open-source tool https://github.com/ChenghaoMou/text-dedup for implementing it.
>
> **Q5**. Line 734: remove deduplicate → deduplicate?
> - **Reply to Q5**. Thank you for your carefully reading. We will refine it in the revised version of our paper. We will also carefully read and check the possible typos in our paper.

---

> > ### Comment · Reviewer_rcgh · 2024-08-08
> >
> > Thanks to the authors for your high-quality clarification! Below are my follow-up comments:
> >
> > **W1**: Your extensive experiments resolve my doubts in W1. However, it’s a pity to see that your “pre-training” data can not fully substitute (DeepSeekCoder-Base-v1.5 + PT SFT << DeepSeekMath-Base + SFT) nor bring significant improvement over SFT (DeepSeekMath-Base + PT SFT improves marginally over DeepSeekMath-Base + SFT) based on continual pre-training like DeepSeekMath. So I would still take this as a limitation for **failing to effectively simplify / push forward the upper limit of the standard pipeline**.
> >
> > P.S. I do not see any response to Q1, which is annotated as replied to. Do I have a misunderstanding?
> >
> > **W2**: I take **experiments in Figure 4 more of showing the importance of selecting high-quality data, instead of “the scaling behavior of the KD dataset size” (of similar quality)**. It would be helpful to **modify the size of the whole raw KD dataset**, select the high-quality data subset in a similar way and compare the results.
> >
> > **W3**: Because we are focusing on the effect on masking in continual pre-training, I want to check whether masking is applied to SFT consistently for the setting “w/o Mask”. If so, the results should resolve my doubts.
> >
> > **W4**: The comparison might be slightly unfair because **KPMath only conducts one-stage SFT on KPMath-Plus**, while you keep the number of the “pre-training” samples the same as KPMath-Plus and conduct **two-stage training, with the total number of your training samples more than KPMath-Plus**. If possible, I would recommend showing the scaling behavior of the final SFT model performance against your training data amount for more comprehensive analysis.
> >
> > **Q3**: Your data selection method using the reward model seems to ignore the prompt information, which might be unreasonable, but still shows competitive performance on MATH and ASDiv. It might be meaningful to try to control the prompt distribution when adopting the “response reward model”.
> >
> > **Minor question**:
> >
> > - Is the “Pre-training Data Proportion” calculated with token or sample?
> >
> > **Typo**:
> >
> > - Line 213: “4.6B math problem-solution pairs” → “… math problem-solution pairs of 4.6B tokens in total”

---

> ### Author Response · Authors · 2024-08-10
> **Reply to Official Comment by Reviewer rcgh**
>
> We sincerely thank the reviewer for the positive feedback and new insightful comments. We list our new responses for W1, P.S. and W2 below:
>
> **W1**: Your extensive experiments resolve my doubts in W1. However, it’s a pity to see that your “pre-training” data can not fully substitute (DeepSeekCoder-Base-v1.5 + PT SFT << DeepSeekMath-Base + SFT) nor bring significant improvement over SFT (DeepSeekMath-Base + PT SFT improves marginally over DeepSeekMath-Base + SFT) based on continual pre-training like DeepSeekMath. So I would still take this as a limitation for failing to effectively simplify / push forward the upper limit of the standard pipeline.
>
> - **Reply to W1**: In this work, we do not focus on replacing the large-scale pre-training stage by our method, but seeking new low-cost approaches for learning mathematical reasoning capability. Our experiments have verified that based on strong LLMs (e.g., Mistral-7B and LLaMA-3), our method can achieve new SOTA performance, enabling them to surpass DeepSeekMath-RL. For DeepSeekCoder-base-v1.5, as it is a relatively weaker code-based LLM, it can not fully elicit the potential of our method.
> Actually, as our approach utilizes DeepSeekMath for synthesizing the pre-training corpus, our synthetic data might be full of its well-learned knowledge. Thus, it might not provide enough new knowledge for greatly improving itself. In contrast, for other models like Mistral-7B and LLaMA-3-8B, our pre-training data can boost the performance a lot. We show the performance in the following table.
>
> |         | GSM8k | MATH-OAI* |	SVAMP	| ASDiv | MAWPS | CARP |
> |------------|------------|------|---------|------------|------------|------|
> | Mistral-7B + SFT	|87.5|	45.8|	87.9|	89.7	|97|	44.4|
> | Mistral-7B + PT + SFT |	88.6	|52.8|	90.4	|92.6	|97.3|	51|
> | LLaMA-3-8B + SFT|	87.3	|45.8	|86.9	|89.1	|96.8	|44|
> | LLaMA-3-8B + PT + SFT|	88.6	|51	|89.4	|92.6	|97.1	|50.9|
>
> **P.S**. I do not see any response to Q1, which is annotated as replied to. Do I have a misunderstanding?
>
> - **Reply to P.S**: We have merged the response to Q1 into Reply to W1-b & Q1, and we also add new experimental results here to better discuss the self-improvement/weak-to-strong generalization.
> For self-improvement, we conduct the experiments on DeepSeekMath, as we use it for data synthesis. As the results shown in the following table, our approach can further boost DeepSeekMath, surpassing DeepSeekMath-RL. It indicates the effectiveness of our approach for self-improvement.
> For weak-to-strong generalization, we utilize our data to pre-train and fine-tune Mixtral-8X7B, the SOTA LLM with Mixture-of-Expert Architecture. As the shown results, our approach can also greatly boost its performance, indicating its effectiveness for weak-to-strong generalization.
>
> |         | GSM8k | MATH |	SVAMP	| ASDiv | MAWPS | CARP |
> |------------|------------|------|---------|------------|------------|------|
> | DeepSeekMath-Base	|64.1|	34.2|	73.7	|82.7	|92.7|	44.4|
> | DeepSeekMath-RL | 88.2 | 50.2 | 87.3 | 91.8 | 95.5 | 51.6 |
> | DeepSeekMath-Base + Ours	|87.6|	52.6	|88.6|	93.1|	96.7|	52|
> | Mixtral-8X7B | 74.4 | 29.0 | 76.5 | 78.5 | 93.9 | 38.8 |
> | Mixtral-8X7B + Ours | 89.8 | 53.8 | 90.2 | 93.1 | 96.7 | 52.3|
>
> **W2**: I take experiments in Figure 4 more of showing the importance of selecting high-quality data, instead of “the scaling behavior of the KD dataset size” (of similar quality). It would be helpful to modify the size of the whole raw KD dataset, select the high-quality data subset in a similar way and compare the results.
>
> - **Reply to W2**: Following the suggestion of the reviewer, we modify the size of the raw dataset to be 100k, 200k, 500k, 1M for scaling the KD dataset amount into 5k, 10k, 25k, and 50k. In this way, we keep the sampling rate of high-quality data to be 5%, to control the similar data quality. We utilize the new KD datasets to fine-tune the data synthesis models. Next, we utilize these models to synthesize top-ranked 100k samples for pre-training Mistral-7B, and then fine-tune on 50k instructions. We report the results on the following table, where all the models achieve similar performance and more data leads to marginal improvement in part of evaluation datasets. It indicates that 5k high-quality KD data is sufficient for training the data synthesis model, and scaling KD data can not greatly improve the performance.
>
> |         | GSM8k | MATH |	ASDiv | CARP |
> |------------|-----------|--------|------------|------------|
> | Ours with 5k KD data | 78.6 | 32.8 | 84.5 | 36.2 |
> | Ours with 10k KD data | 78.8 | 32.7 | 84.1 | 36.3 |
> | Ours with 25k KD data | 78.9 | 33.0 | 84.6 | 35.9 |
> | Ours with 50k KD data | 78.8 | 32.8 | 84.5 | 36.5 |

---

> ### Author Response · Authors · 2024-08-10
> **Reply to Official Comment by Reviewer rcgh (part-2)**
>
> We sincerely thank the reviewer for the positive feedback and new insightful comments. We list our new responses for W3, W4, Q3, Minor Question and Typo below:
>
> **W3**: Because we are focusing on the effect on masking in continual pre-training, I want to check whether masking is applied to SFT consistently for the setting “w/o Mask”. If so, the results should resolve my doubts.
>
> - **Reply to W3**: Yes, masking is applied in the SFT process, following the classic settings in existing work[1,2]. In the above experiments, we only use or remove the Mask in the PT process.
>
> [1] Long Long Yu, Weisen Jiang, Han Shi, Jincheng Yu, Zhengying Liu, Yu Zhang, James T. Kwok, Zheng Li, Adrian Weller, and Weiyang Liu. Metamath: Bootstrap your own mathematical questions for large language models. ArXiv, abs/2309.12284, 2023
>
> [2] Huaiyuan Ying, Shuo Zhang, Linyang Li, Zhejian Zhou, Yunfan Shao, Zhaoye Fei, Yichuan Ma, Jiawei Hong, Kuikun Liu, Ziyi Wang, Yudong Wang, Zijian Wu, Shuaibin Li, Fengzhe Zhou, Hongwei Liu, Songyang Zhang, Wenwei Zhang, Hang Yan, Xipeng Qiu, Jiayu Wang, Kai Chen, and Dahua Lin. Internlm-math: Open math large language models toward verifiable reasoning. ArXiv, abs/2402.06332, 2024
>
> **W4**: The comparison might be slightly unfair because KPMath only conducts one-stage SFT on KPMath-Plus, while you keep the number of the “pre-training” samples the same as KPMath-Plus and conduct two-stage training, with the total number of your training samples more than KPMath-Plus. If possible, I would recommend showing the scaling behavior of the final SFT model performance against your training data amount for more comprehensive analysis.
>
> - **Reply to W4**: Actually, we guarantee the total number of training samples for our method (including PT+SFT data) and KPMath-Plus to be the same. Thus, according to the results, we can compare the effectiveness of our approach and KPMath-Plus, with the same training cost.
> Following the suggestion of the reviewer, we also scale the amount of pre-training data from 500k, 1M, 2M and 6M, and control the SFT data amount unchanged. We report the final model performance in the following table, where we can see that the performance increases w.r.t. the increasing of pre-training data amount.
>
> |         | GSM8k | MATH |	SVAMP	| ASDiv |
> |------------|------------|------|---------|------------|
> | Mistral + 500k CoT PT + All SFT |	88.2|	49|	87.9|	90.9|
> | Mistral + 1M CoT PT + All SFT |	87.9	|49|	89.6|	91.3|
> | Mistral + 2M CoT PT + All SFT | 88.4	 | 50.8	|89.9|	91.4|
> | Mistral + 6M CoT PT + All SFT |	88.6	|52.8|	90.4|	92.6|
>
> **Q3**: Your data selection method using the reward model seems to ignore the prompt information, which might be unreasonable, but still shows competitive performance on MATH and ASDiv. It might be meaningful to try to control the prompt distribution when adopting the “response reward model”.
>
> - **Reply to Q3**: In fact, the Reward Model does not ignore prompts, as it measures whether the response has well solved the question in the prompt. It is a rather useful aspect to estimate the quality of the whole sample, and enables to identify more helpful samples for training.
> Following the suggestion of the reviewer, we also conduct the experiment to compare the selected samples with or without controling the prompt distribution when adopting the reward model. Concretely, we control that the rates of selected samples for different prompts are the same, i.e., 5%. We find that there is a large overlap between the selected samples from the two variations, nearly 95%. Also, the selected samples using different prompts by our original method naturally own similar proportion as the original dataset. Both indicate that it is unnecessary to control the prompt distribution during data selection.
>
> **Minor Question**: Is the “Pre-training Data Proportion” calculated with token or sample?
>
> - **Reply to Minor Question**: It refers to the proportion of samples, as it is hard to accurately control the token number for matching the selected data proportion.
>
> **Typo**: Line 213: “4.6B math problem-solution pairs” → “… math problem-solution pairs of 4.6B tokens in total”
>
> - **Reply to Typo**: Thank you for your carefully reading. We will refine it in the revised version of our paper.

---

> > ### Comment · Reviewer_rcgh · 2024-08-10
> >
> > Thanks for your high-quality reply! You have almost resolved all my concerns!
> >
> > But sorry that I **would not like to raise my score >= 8 if a method could not be effectively justified to be able to simplify / improve the SotA pipeline**. Though results based on Mistral-7B&Llama-3-8B are impressive, it is highly possible that combining DeepSeekMath and JiuZhang3.0 could achieve better results but not clear if JiuZhang3.0 is truly necessary. The only way that I come up with for now to validate this is to conduct ablation studies on DeepSeekCoder-base-v1.5, but if you would not like to explore this, it would be fine.
> >
> > Considering the results so far, I would keep my score as 7. However, your dedicated discussion indeed makes the submission of a higher quality. Again, thanks for your efforts in the discussion!

---

> > > ### Author Response · Authors · 2024-08-11
> > > **New SOTA Results by Scaling our Synthetic Data into 10B**
> > >
> > > Thank you for your positive feedback and acknowledgment of our effort put into addressing your concerns. Actually, we will appreciate it if you would consider raising the score after seeing our new SOTA experimental results.
> > >
> > > Inspired by your concern about if our method can simplify or enhance the SOTA pipeline, we carefully rethink all the settings in our method. Surprisely, we find that due to our limitation on the pre-training data amount (4.6B), we have not even reached the effectiveness ceiling of our method. Concretely, we scale up the amount of the synthetic pre-training data from 4.6B to 10B, and utilize it to train DeepSeekCoder-v1.5 and Mistral-7B then fine-tuning on the original SFT data. We report the results in the following table, where a great performance improvement can be observed in the two LLMs.
> > >
> > > Firstly, more pre-training data helps DeepSeekCoder-Base-v1.5 outperform DeepSeekMath-RL on all the evaluation datasets. It indicates that we achieve the goal of using rather less data (10B) to reach the performance of SOTA large-scale pre-trained LLMs (DeepSeekMath using 120B data).
> > >
> > > Second, more pre-training data improves the performance of Mistral-7B a lot. We  achieve new SOTA among open-source LLMs, and greatly narrow the performance gap between open-source LLMs and closed-source GPT-4.
> > >
> > > As we will publicly release all the synthetic data, we believe our method would be a promising solution to simplify the SOTA pipeline. Everyone can utilize our data to efficiently pre-train the LLM, which greatly reduces the data amount requirement and can achieve better performance. With more computing resource, it is also promising to further scale up the synthetic dataset or fine-tune stronger LLMs as the data synthetic model.
> > >
> > > |         | GSM8k | MATH |	SVAMP	| ASDiv | MAWPS | CARP |
> > > |------------|------------|------|---------|------------|------------|------|
> > > | DeepSeekMath-RL | 88.2 | 50.2 | 87.3 | 91.8 | 95.5 | 51.6 |
> > > | DeepSeekCoder-Base-v1.5 + 4.6B-PT + SFT|86.2|	47.4|87.2|90.9|	96.8|	49.6|
> > > | DeepSeekCoder-Base-v1.5 + 10B-PT + SFT|88.4 | 53.0|88.6 |92.6|97.6|51.7 |
> > > | Mistral-7B + 4.6B-PT + SFT |	88.6	|52.8|	90.4	|92.6	|97.3|	51|
> > > | Mistral-7B + 10B-PT + SFT |	90.4	|56.2|	91.1	|93.3	|97.1|	51.1|
> > > | GPT-4 | 92.2 | 65.4 | 92.9 | 94.3 | 96.6 | 53.6 |

---

> > > > ### Comment · Reviewer_rcgh · 2024-08-11
> > > >
> > > > Thanks for your reply! Your newest results are really impressive!
> > > >
> > > > My last concern is **the potential generalization issue introduced by LESS with in-domain samples of the test tasks**. Could you test the SotA models and their most comparable baselines on some out-of-domain tasks, e.g. TheoremQA, MWPBench/CollegeMath/Test, MWPBench/GaokaoBench, Odyssey-Math?
> > > >
> > > > I would consider improving my rating to 8/9 based on the generalization results.

---

> ### Author Response · Authors · 2024-08-12
> **New Experiments on 2 Settings and 17 Datasets for Testing Generalization Capability**
>
> We sincerely thank the reviewer for the positive feedback and the patient about our discussion. Following the suggestion, we test the generalization capability of our newly trained LLMs in two settings, i.e., natural language reasoning and tool manipulation, totally  17 datasets. We report the results of our methods and baselines in the following two tables, where the first table is for natural language reasoning and the second is for tool manipulation settings, respectively.
>
> First, with 10B PT data, DeepSeekCoder can perform better than DeepSeekMath-7B-RL on most evaluation datasets from the two settings. It indicates that the scaling of pre-training data is an effective approach to improving the general capability of LLMs.
>
> Besides, 10B PT data also leads to great performance gain on Mistral-7B, achieving new SOTA on all the datasets. It further demonstrates that our method is a promising solution to simplify the SOTA pipeline.
>
>
> |           | MWPBench/Collge Math                  | MWPBench/GaokaoBench | MWPBench/Fresh Gaokao 2023 | Gaokao2023 Math En | OlympiadBench | DeepMind Mathematics | GSM Plus | Math Odyssey | TheoremQA | Minerva Math |
> |---------------------------------------|----------------------|----------------------------|--------------------|---------------|----------------------|----------|--------------|-----------|--------------|-------|
> | DeepSeekMath-7B-RL                   | 37.5                 | 42.7                       | 36.7               | 43.6          | 19.3                 | 57.3     | 66.7         | **36.2**      | **37.9**         | 20.6  |
> | DeepSeekCoder-7B-base + 10B PT + SFT | 37.8                 | 40.7                       | 40.0               | 48.1          | 20.0                 | 60.0     | 67.1         | 30.5      | 26.9         | 21.0  |
> | Mistral 7B + 4.6B PT + SFT                        | 38.2                 | 38.8                       | 43.3               | 48.8          | 22.2                 | 57.2     | 68.4         | 33.9      | 29.3         | 20.2  |
> | Mistral 7B + 10B PT + SFT             | **41.7**                 | **46.1**                       | **56.7**               | **51.2**          | **25.6**                 | **67.0**     | **71.2**         | 34.4      | 32.5        | **25.4**  |
>
> |   | GSM8K                                 | MATH  | GSM Hard | SVAMP   | TabMWP | ASDiv  | MAWPS  |
> |---------------------------------------|-------|----------|---------|--------|--------|--------|-------|
> | DeepSeekMath-7b-RL           | 78.5  | 52.8     | 61.3    | 85.9   | **80.3**   | 87.0   | 95.8  |
> | DeepSeekCoder-7b + 10B PT + SFT | 81.7  | 52.0     | 64.7    | 87.9   | 78.5   | 89.8   | 96.9  |
> | Mistral 7B + 4.6B PT + SFT                        | 82.4  | 53.0     | **64.9**    | 89.2   | 75.6   | 88.3   | 96.6  |
> | Mistral 7B + 10B PT + SFT             | **91.1**  | **54.8**     | 61.2    | **91.3**   | 79.6   | **93.4**   | **97.4**  |
>
> After we publicly release our data, everyone can utilize it to pre-train the LLM, which greatly reduces the data amount requirement and achieve exciting performance. As the rebuttal DDL is approaching, we have to leave more interesting investigation about testing the ceiling of our approach into future work, including further scaling up the synthetic dataset or fine-tuning stronger LLMs for data synthesis.
>
> Thus, we sincerely appreciate it if you would raise the score, as it is indeed very important for our work.

---

> > ### Comment · Reviewer_rcgh · 2024-08-12
> >
> > Thanks for your reply! It is also happy for me to see the impressive results. I will update my official review.

---

> > > ### Author Response · Authors · 2024-08-12
> > > **Sincerely Thank for Reviewer rcgh**
> > >
> > > Dear Reviewer rcgh,
> > >
> > > We sincerely thank you for the updated score.
> > >
> > > We appreciate your valuable feedback, which helped enhance our paper significantly. We will revise our paper according to your valuable suggestions! For your further comments, they are also constructive suggestions and we will study them in the future work.
> > >
> > > Thank you once again for taking the time to review our paper and discussing with us!
> > >
> > > Best,
> > >
> > > Authors

---

### Official Review · Reviewer_trb5 · 2024-07-15

**Soundness:** 3
**Presentation:** 3
**Contribution:** 3
**Rating:** 7
**Confidence:** 4

**Summary:**

This work proposes a low-cost yet efficient method to synthesize QA pairs using a small LLM (e.g., DeepSeek-Math-7B model) instead of GPT-4, significantly reducing cost and improving performance in mathematical reasoning.

Concretely, it begins by initializing the data synthesis LLM through distillation from GPT-4 using randomly sampled data. It then enhances the model using high-value data selected via a gradient-based value estimation strategy and regenerates QA pairs from GPT-4, ultimately synthesizing around 6M QA pairs on diverse math pretraining corpus.

The synthetic QA pairs are applied to pretrain open-source LLMs, which are then fine-tuned on collected SFT data, resulting in improved performances across benchmarks such as GSM8K and MATH.

**Strengths:**

- The feasibility of training a 7B LLM to replace GPT-4 for synthesizing QA pairs for mathematical pretraining is demonstrated.
- The cost is significantly lower than previous state-of-the-art methods (e.g., Deepseek-Math).
- The final performance, after pretraining and SFT stages, surpasses state-of-the-art methods in both natural language reasoning and tool manipulation settings.
- Extensive ablation studies confirm the effectiveness of their pipeline.

**Weaknesses:**

- The core idea involves distilling GPT-4’s synthesis capability into a smaller yet robust 7B LLM (e.g., Deepseek-Math-7B), which remains within the scope of distilling GPT-4 to open LLMs. The pipeline is somewhat compositional and complex, especially concerning the gradient-based value estimation.
- The main experimental result (Table 1) only displays the final performance, omitting a comparison of pretraining performance. This omission may lead to confusion about whether improvements result from the pretraining or SFT data.
- The work focuses on synthesizing QA pairs. The validity of the questions and the correctness of the solutions are crucial aspects of this line of work. This work does not address these aspects but only provides several examples in Appendix I.

**Questions:**

- Regarding Weakness 2: Is there a fair comparison with Deepseek-Math-7B-Base in terms of pretraining performance in Table 1?
- Regarding Weakness 3: How do you consider the validity of the questions and the correctness of the solutions in your pipeline?
- Fine-tuning Data: In Appendix D, you mention using a unified prompt for DeepSeekMath-7B-RL to synthesize 700k solutions for problems from your instruction tuning datasets. Does this mean that no original solutions are used in your fine-tuning data, potentially losing useful data points? For example, MetaMath introduces augmented answers for GSM8K and MATH and shows benefits.
- Data Synthesis Models: In Appendix G, you mention initializing the data synthesis models with DeepSeekMath-7B-RL. Why choose DeepSeekMath-7B-RL rather than DeepSeekMath-7B-Base? Is there any experimental comparison?

**Limitations:**

See Weaknesses

---

> ### Author Rebuttal · Authors · 2024-08-04
>
> We sincerely thank the reviewer for the insightful suggestion and positive feedback, and list our responses for the weaknesses below:
>
> **W1**. The core idea involves distilling GPT-4’s synthesis capability into a smaller yet robust 7B LLM (e.g., Deepseek-Math-7B), which remains within the scope of distilling GPT-4 to open LLMs. The pipeline is somewhat compositional and complex, especially concerning the gradient-based value estimation.
> - **Reply to W1**. In this work, we aim to verify whether we can improve mathematical reasoning capability of LLMs via a low-cost way. Thus, we need to distill GPT-4's synthesis capability into a small LLM for low-cost massive math problems synthesis. However, it is not easy to achieve this goal, as math problems are typically complex and diverse. Besides, we also should control the cost of invoking GPT-4 APIs to be acceptable. To this end, we expect to first select less high-value data, and then only use GPT-4 to create knowledge distillation dataset based on it. Therefore, we adopt the gradient-based value estimation method for measuring and selecting the high-value data. In our empirical experiments and the ablation study in Table 4, we can see that all the components are useful. Without the value estimation method, the LLM suffers performance degradation. Note that as we carefully design the pipeline to train the data synthesis model, it can be easily used for data synthesis, with no need for retraining the model. Also, our synthetic dataset can also be directly used to pre-train LLMs for improving their mathematical reasoning capability.
>
> **W2 & Q1**. The main experimental result (Table 1) only displays the final performance, omitting a comparison of pretraining performance. This omission may lead to confusion about whether improvements result from the pretraining or SFT data.
> - **Reply to W2 & Q1**. Following the suggestion of the reviewer, we train Mistral-7B only using the pre-trained or SFT data, respectively, and also report the performance of DeepSeekMath-Base and JiuZhang3.0-7B for comparison. As shown in the following Table, after using the pre-training data, our base model can outperform DeepSeekMath-Base, SOTA open-source backbone for mathematical reasoning, indicating the effectiveness of our pre-training data. Besides, by only using the SFT data, Mistral-7B+SFT underperforms JiuZhang3.0-7B in a large margin. It also demonstrates that our pre-training data can improve the potential of LLMs for learning the SFT data.
>
> |         | GSM8k | MATH |	SVAMP	| ASDiv | MAWPS | CARP |
> |------------|------------|------|---------|------------|------------|------|
> | DeepSeekMath-Base-7B	|64.1|	34.2|	73.7	|82.7	|92.7|	44.4|
> | JiuZhang3.0-7B-Base|	68.7	|40.2	|80.4	|86.5	|93.5	|40.2|
> | Mistral-7B + SFT	|87.5|	45.8|	87.9|	89.7	|97|	44.4|
> | JiuZhang3.0-7B |	88.6	|52.8|	90.4	|92.6	|97.3|	51|
>
>
> **W3 & Q2**. The work focuses on synthesizing QA pairs. The validity of the questions and the correctness of the solutions are crucial aspects of this line of work. This work does not address these aspects but only provides several examples in Appendix I.
> - **Reply to W3 & Q2**. In existing work[1,2], it is inevitable that erroneous and noisy samples are included into the pre-training data of LLMs. Whereas, its negative influence can be greatly reduced by the following supervised-finetuning process, using correct and human-aligned data[1,3]. For mathematical reasoning, our empirical experiments also found that LLMs are robust to the noise during pre-training. Concretely, we utilize a well-trained LLM to annotate the accuracy of all the pre-training data, then only select the top-ranked 500k samples for pre-training. Besides, we also randomly sample 500k samples for pre-training, to compare LLMs pre-training on the original or correct data. After pre-training, we fine-tune the two models on 50k instructions. As shown in the following table, the two variations achieve comparable performance, indicating that the noise is not an important concern.
>
> [1] Ouyang, Long, et al. "Training language models to follow instructions with human feedback." Advances in neural information processing systems 35 (2022): 27730-27744.
>
> [2] Albalak, Alon, et al. "A survey on data selection for language models." arXiv preprint arXiv:2402.16827 (2024).
>
> [3] Zhou, Chunting, et al. "Lima: Less is more for alignment." Advances in Neural Information Processing Systems 36 (2024).
>
> |         | GSM8k | MATH |	SVAMP	| ASDiv | MAWPS | CARP | Avg. |
> |------------|------------|------|---------|------------|------------|------|---------|
> | **PT Random 500k + SFT 50k** | 83.2 | 38.4 | 84.9 | 88.3 | 94.8 | 43.6 | 72.2 |
> | **PT Top-Acc. 500k + SFT 50k** | 82.7 | 40.8 | 84.1 | 87.9 | 94.2 | 42.2 | 72.0 |

---

> ### Author Response · Authors · 2024-08-04
> **Rebuttal by Authors (part-2)**
>
> We sincerely thank the reviewer for the insightful suggestion and positive feedback, and list our responses for the questions below:
>
> **Q3**. Fine-tuning Data: In Appendix D, you mention using a unified prompt for DeepSeekMath-7B-RL to synthesize 700k solutions for problems from your instruction tuning datasets. Does this mean that no original solutions are used in your fine-tuning data, potentially losing useful data points? For example, MetaMath introduces augmented answers for GSM8K and MATH and shows benefits.
> - **Reply to Q3**. We use the original solutions of GSM8k and MATH in our fine-tuning data, as they are human-annotated high-quality data. In addition to it, we also use a unified prompt for DeepSeekMath-7B-RL to synthesize 700k solutions, which help regulate the output format of LLMs, to avoid misleading by diverse formats from collected instruction datasets.
>
> **Q4**. Data Synthesis Models: In Appendix G, you mention initializing the data synthesis models with DeepSeekMath-7B-RL. Why choose DeepSeekMath-7B-RL rather than DeepSeekMath-7B-Base? Is there any experimental comparison?
> - **Reply to Q4**. According to the original paper, DeepSeekMath-7B-RL is trained with more instructions and math questions, it also performs much better than DeepSeekMath-7B-base. Thus, it would own better math reasoning capability than the base model. Besides, we have also conducted experiments to study which one is a better data synthesis model. Concretely, we replace DeepSeekMath-7B-RL by DeepSeekMath-7B-base in our pipeline, and utilize them to synthesize the same number of samples for pre-training. Then, we pre-train Mistral-7B on the synthetic samples, and then fine-tune on our collected SFT data. As shown in the following Table, using DeepSeekMath-7B-RL can achieve slightly better performance than using DeepSeekMath-7B-base. Thus, we choose DeepSeekMath-7B-RL in our work.
>
> |         | GSM8k | MATH |	SVAMP	| ASDiv | MAWPS | CARP |
> |------------|------------|------|---------|------------|------------|------|
> | **Ours + DeepSeekMath-7B-base** 	|89 |	52.6	|89.9|	91.4|	97.3|	50.4|
> | **Ours + DeepSeekMath-7B-RL** |	88.6	|52.8|	90.4	|92.6	|97.3|	51|

---

> ### Author Response · Authors · 2024-08-12
> **Urgent Reminder of Rebuttal**
>
> Dear Reviewer trb5,
>
> Sorry to disturb you for the last time, but only one day is left until the end of the reviewer-author discussion stage. We still do not know if you have received our response. To address your concerns, we wrote all the responses in details and added new experiments to support it, including:
> - (1) clarify our contributions in Reply to W1;
> - (2) train Mistral-7B only using the pre-trained or SFT data in Reply to W2 & Q1;
> - (3) empirically validate that LLMs are robust to correctness noise in Reply to W3 & Q2;
> - (4) clarify the used prompt for the SFT data in Reply to Q3;
> - (5) train our DeepSeekMath-7B-base as data synthesis model in Reply to Q4;
>
> You know, conducting the above mentioned experiments within the limited rebuttal period was quite challenging. We would like to know whether our responses have addressed your concerns. If you still have other concerns, please give us an opportunity to clarify them.
>
> Sincerely hope that you can take a moment to reply to us, as it is very important for researchers and their efforts on this work.
>
> Best regards,
>
> The Authors

---

> ### Comment · Reviewer_trb5 · 2024-08-13
> **thanks**
>
> I would like to keep the score as positive.

---

> > ### Author Response · Authors · 2024-08-13
> > **Sincerely Thank for Reviewer trb5**
> >
> > Dear Reviewer trb5,
> >
> > We sincerely thank you for your response and valuable feedback that helped enhance our paper significantly. If you have any additional queries or concerns about preventing you from raising your score, please don't hesitate to inform us, and we'll do our best to respond promptly.
> >
> > Thank you once again for taking the time to review our paper and read our comments.
> >
> > Best,
> >
> > Authors

---

### Official Review · Reviewer_FBVa · 2024-07-18

**Soundness:** 3
**Presentation:** 3
**Contribution:** 2
**Rating:** 5
**Confidence:** 3

**Summary:**

This paper proposes a framework for training large-scale mathematical reasoning models using synthetic data, introducing JiuZhang3.0 based on this framework. Compared to manually collected or LLM-generated training data, this framework significantly reduces resource costs. The authors trained three different scales of JiuZhang3.0 models based on Mistral-7B, LLaMa-3-8B, and Mixtral-8*7B, demonstrating the effectiveness of this framework through extensive experiments.

**Strengths:**

1. Proposed an effective training framework that can generate a large amount of high-quality training data with only a few GPT-4 calls.
2. This work compares various open-source (Qwen series, Mixtral-8*7B) and closed-source models (ChatGPT, GPT-4), and conducts experiments on 18 datasets including GSM8k and MATH, with very rich experimental results.

**Weaknesses:**

1. Novelty: Overall, I believe the innovation of this work is limited. The core of this work is a framework for generating high-quality training data. However, using GPT-4 to generate data to fine-tune smaller models is not innovative, as shown in [1]. Additionally, in the Boost Synthesis LLM stage, the work essentially just uses the methods from Xia [2] et al., lacking its own innovation. This framework seems more like a fusion of multiple works.
2. The authors conducted many ablation experiments, but there is a lack of in-depth analysis. For example, why is directly using GPT-4 to generate questions and answers not as effective as the original method? In the choice of Synthesis LLM, why is there such a large discrepancy? I find the authors' explanations vague.
3. There are some minor writing flaws worth noting, such as the uniformity of m_l and the capitalization in equation 3 on line 163.

**Questions:**

1. Regarding the title. I think the contribution of this paper is a framework that can generate high-quality training data, not the model itself. The title JiuZhang3.0 may mislead readers into thinking the core contribution is the model. I don't understand why the authors chose this title.
2. It seems that Initialize Synthesis LLM and Boost Synthesis LLM can be iterative. Can multiple iterations train a stronger Data Synthesis Model? Would this help improve the performance of JiuZhang3.0?
3. Regarding lines 135-136, the math-related texts used by the authors. Could the authors provide a more detailed explanation of how these texts were obtained? The ablation experiment results indicate it has a significant impact on the overall framework, so I think it is necessary to provide a detailed explanation.

**Limitations:**

Yes. The authors adequately addressed the limitations.

---

> ### Author Rebuttal · Authors · 2024-08-04
>
> We sincerely thank the reviewer for the insightful suggestion, and list our responses for the weaknesses below:
>
> **W1**. Novelty: Overall, I believe the innovation of this work is limited. The core of this work is a framework for generating high-quality training data. However, using GPT-4 to generate data to fine-tune smaller models is not innovative, as shown in [1]. Additionally, in the Boost Synthesis LLM stage, the work essentially just uses the methods from Xia [2] et al., lacking its own innovation. This framework seems more like a fusion of multiple works.
> - **Reply to W1 - Part1**. In this work, we aim to verify whether we can efficiently improve mathematical reasoning capability of LLMs by training small data synthesis models. As the reviewer mentioned, existing work has used GPT-4 to generate data for fine-tuning LLMs to improve their mathematical reasoning capability. However, it needs GPT-4 to generate a number of samples for achieving SOTA performance (e.g., KPMath-DSMath-7B[1] using GPT-4 865k times), which is not an efficient way and approximately spends 40k USD. In this work, we train a small LLM specially for data synthesis, and only need GPT-4 to synthesize few samples (i.e., 10k) for distilling the data synthesis capability into it. Our work demonstrates that such a way is applicable and can achieve new SOTA performance with only 20% cost (i.e., 8.5k USD), shown in Figure 1. Thus, our work is promising to inspire other researchers for exploring how to further reduce the cost for LLM to learn new capabilities, advancing the development of low-carbon LLMs and green AI.
> - **Reply to W1 - Part2**. Besides, the major technical contribution of this work is that we propose a low-cost solution for distilling the data synthesis capability from GPT-4 into small LLMs. To this end, we craft a set of prompts corresponding to human education stages for ensuring the knowledge coverage, and adopt an influence estimation strategy to select high-value math-related texts for feeding into GPT-4. The influence estimation strategy is indeed borrowed from existing work, but we find that the original method can not be directly applied into our approach in an efficient way. Therefore, we devise the following optimizations: (1) applying the selection method on math-related texts instead of GPT-4 synthetic data to reduce the cost; (2) adding the first stage for initializing synthesis LLM to guarantee the quality of candidates. In this way, we can ensure both the effectiveness and low cost of the influence estimation strategy in our work.
>
> **W2**. The authors conducted many ablation experiments, but there is a lack of in-depth analysis. For example, why is directly using GPT-4 to generate questions and answers not as effective as the original method? In the choice of Synthesis LLM, why is there such a large discrepancy? I find the authors' explanations vague.
> - **Reply to W2 - Part1**. Thank you for the constructive comment. Due to the page limit, we do not deeply analyze the experimental results. Here, we will do it and add the analysis in the revised version.
> For ablation study, the performance of the variations that remove prompt set or math-related texts degrades a lot. It is because the prompts can control the synthetic samples corresponding to human educational stages, and the math-related texts contain rich math domain knowledge. Both are important to guarantee the knowledge coverage and quality of the synthetic data. Besides, without using GPT-4 for boosting, the variation degrades a lot in MATH. The reason is that it is not easy for the small LLM itself to synthesize complex math problems that are helpful for the MATH dataset, without the help of GPT-4.
> - **Reply to W2 - Part2**. For synthesis LLM, we select the four models owing to their better evaluation performance in Table 1,2,3, which implies that they own better math reasoning capability and may perform well on math problem synthesis. Among them, the variation direcly using ChatGPT performs not well. A possible reason is that ChatGPT is not specially trained for mathematical reasoning, its capability of math data synthesis is relatively weak. Furthermore, DeepSeekMath-RL-7B also performs not better than our approach. It indicates that distilling the data synthesis capability from GPT-4 is necessary for adapting math-related LLMs into proper data synthesis model.
> - **Reply to W2 - Part3**. For data selection, the variation using perplexity does not perform well. According to our case study, we find that the samples with lower perplexity are typically easier ones, which are not very helpful to learn to solve complex math problems. Additionally, the variation using one-hot ICL performs relatively better. We find the ICL loss are prone to assign high values for the solutions with more natural language tokens, which is more useful for GSM8k but not good for MATH with more math symbols and formulas.
>
> **W3**. There are some minor writing flaws worth noting, such as the uniformity of m_l and the capitalization in equation 3 on line 163.
> - **Reply to W3**. Thank you for your carefully reading. We will fix your mentioned writing flaws, and carefully check and refine the writing of our paper in the revised version.

---

> > ### Comment · Reviewer_FBVa · 2024-08-12
> > **Reviewer Response**
> >
> > I have read the response. Thanks for your clarification.

---

> > > ### Author Response · Authors · 2024-08-12
> > > **Sincerely Thank for Reviewer FBVa**
> > >
> > > Dear Reviewer FBVa,
> > >
> > > We sincerely thank you for the updated score, and appreciate your valuable feedback. We will revise our paper according to your valuable suggestions!
> > >
> > > Thank you for taking the time to review our paper and read our comments. If you have any additional queries or concerns, please don't hesitate to inform us, and we'll do our best to respond promptly.
> > >
> > > Best,
> > >
> > > Authors

---

> ### Author Response · Authors · 2024-08-04
> **Rebuttal by Authors (part-2)**
>
> We sincerely thank the reviewer for the insightful suggestion, and list our responses for the questions below:
>
> **Q1**. Regarding the title. I think the contribution of this paper is a framework that can generate high-quality training data, not the model itself. The title JiuZhang3.0 may mislead readers into thinking the core contribution is the model. I don't understand why the authors chose this title.
> - **Reply to Q1**. Thank you for your valuable suggestion. This title is named due to our consideration of our JiuZhang series language models for math problem solving. We will change it to avoid possible misunderstanding about this work.
>
> **Q2**. It seems that Initialize Synthesis LLM and Boost Synthesis LLM can be iterative. Can multiple iterations train a stronger Data Synthesis Model? Would this help improve the performance of JiuZhang3.0?
> - **Reply to Q2**. Following the suggestion of the reviewer, we also have conducted the experiments that iterate the process for further boosting the data synthesis model. Concretely, we repeat the 2nd stage that first selects 4k high-value data and then uses it for distilling knowledge from GPT-4 into data synthesis model. We repeat the above process once again, and then synthesizes 100k samples for pre-training. Following the setting in our ablation study, we then fine-tune the LLM on 50k samples. As the results shown in the following table, iteration can not further boost the performance, even causing performance degradation. The reason may be that data synthesis is not a hard-to-learn capability for small LLMs. The original 10k data is sufficient for well training the data synthesis model.
>
> |         | GSM8k | MATH |	ASDiv | CARP |
> |------------|-----------|--------|------------|------------|
> | **Ours** |78.6|	32.8|	84.5|	36.2|
> | **Ours + 1 round Iteration**| 75.4	|31.8|	83.0|	35.9|
>
> **Q3**. Regarding lines 135-136, the math-related texts used by the authors. Could the authors provide a more detailed explanation of how these texts were obtained? The ablation experiment results indicate it has a significant impact on the overall framework, so I think it is necessary to provide a detailed explanation.
> - **Reply to Q3**. Thank you for your constructive suggestion. As written in Section 2.4, we select the commonly-used math-related data from multiple resource, consisting of OpenWebText, Mathpile, StackExchange. OpenWebText is widely-used for pre-training math-related LLMs, which contains the math-related webpages from Common Crawl. Mathpile is a mixture of multi-source datasets, including math-related textbooks, Arxiv papers, and wikipedia, which contain rich math knowledge with diverse styles. StackExchange contains massive real-world math-related questions, which is also very useful for learning to master mathematical knowledge. We will open source all the above data and code, to help researchers reimplement our work.

---

> ### Author Response · Authors · 2024-08-12
> **Urgent Reminder of Rebuttal**
>
> Dear Reviewer FBVa,
>
> Sorry to disturb you for the last time, but only one day is left until the end of the reviewer-author discussion stage. We still do not know if you have received our response. To address your concerns, we wrote all the responses in details and added new experiments to support it, including:
> - (1) clarify our novelty and contributions in Reply to W1;
> - (2) provide in-depth analysis about the ablation study experiments in Reply to W2;
> - (3) conduct experiments about 1-round Iteration on our method in Reply to Q2;
> - (4) other detailed replies and modification for other suggestions;
>
> We would like to know whether our responses have addressed your concerns. If you still have other concerns, please give us an opportunity to clarify them.
>
> Sincerely hope that you can take a moment to reply to us, as it is very important for researchers and their efforts on this work.
>
> Best regards,
>
> The Authors

---

### Official Review · Reviewer_1oiT · 2024-07-24

**Soundness:** 3
**Presentation:** 3
**Contribution:** 3
**Rating:** 5
**Confidence:** 5

**Summary:**

The paper addresses the high cost of training large language models (LLMs) for mathematical reasoning by proposing a more efficient method for generating high-quality pre-training data using a smaller LLM. Instead of relying on extensive pre-training data or using powerful models like GPT-4 for large-scale synthesis, the authors train a small LLM to synthesize math problems. They create a dataset by distilling GPT-4's synthesis capability into the small LLM using carefully crafted prompts covering various educational stages. Additionally, they employ a gradient-based influence estimation method to select the most valuable math-related texts for creating a knowledge distillation dataset. This approach significantly reduces the need for GPT-4 API calls and large-scale pre-training data. The resulting model, JiuZhang3.0, achieves state-of-the-art performance on several mathematical reasoning benchmarks.

**Strengths:**

The proposed methodology for enhancing mathematical reasoning in LLMs through cost-efficient data synthesis has several advantages: 1. Cost Efficiency: Utilizing a smaller LLM for data synthesis drastically reduces training and inference costs, requiring only 9.3k GPT-4 API calls; 2. High-Quality Data: The approach ensures high-quality synthesized data by using carefully crafted prompts and selecting the most valuable math-related texts through gradient-based influence estimation; 3. Scalability: A small LLM can efficiently generate a large dataset (4.6B tokens) for pre-training, making the method scalable; 4. Performance: JiuZhang3.0 achieves state-of-the-art results on various mathematical reasoning datasets, outperforming larger models in natural language reasoning and tool manipulation settings; 5. Generalizability: Demonstrates that smaller models can effectively learn and adapt to complex tasks with proper training strategies, applicable to other domains beyond mathematics. The paper is well-written.

**Weaknesses:**

Some details of the paper are not clear, please refer to the following questions.

**Questions:**

1. Table 1 shows that as the model size increases, the performance does not improve significantly, and some metrics even decline.
2. The accuracy of the pre-trained synthetic data raises concerns about the noise introduced by numerous erroneous samples.
3. Performance comparison of pre-training using original data versus synthetic data.
4. Differences between using gradient descent and semantic similarity methods to filter high-quality data.
5. Both Table 1 and Table 3 indicate that reasoning with CoT and Tool assistance yields similar performance.
6. The loss calculation method for the pre-trained model and its few-shot performance on some benchmarks, like GSM8k and MATH.
7. Figure 3 illustrates that as the data volume increases, the performance of two base models initially rises and then falls, rather than continuously improving.
8. Is fine-tuning Llama3 more suitable for your proposed synthetic method compared to Mistral? Which is more appropriate during the SFT phase: Llama3 or Mistral?
9. Why was 10k selected for training the synthetic model? Could more data be used?
10. Comparing the synthetic data capabilities of different LLMs in Table 4 seems unfair, as the study used distilled GPT-4 for synthetic data generation.
11. The performance did not significantly improve when comparing different data selection strategies in Table 4.
12. The effect of using SFT data on the Mistral-7B base model.
13. There is an error in citation [50].

**Limitations:**

Yes

---

> ### Author Rebuttal · Authors · 2024-08-04
>
> We sincerely thank the reviewer for the insightful suggestion and positive feedback, and list our responses for the weaknesses 1-5 below:
>
> **W1**. Table 1 shows that as the model size increases, the performance does not improve significantly, and some metrics even decline.
> - **Reply to W1**. As shown in existing work (e.g., LLaMA-3-8B and Mistral-7B), small LLMs can also perform pretty well by training on larger high-quality corpus, even outperforming larger LLMs. Therefore, the performance does not always consistently increase w.r.t. the scaling of model size. This phenomenon can also be seen on other baselines in Table 1 (e.g., Qwen-1.5-110B and DeepSeekMath-7B-RL). Moreover, as the used backbones are not trained with the same data, their capacity and compatibility for our approach would also be different. But according to the average performance on Table 1,2,3, they roughly follow the tendency that JiuZhang3.0-8x7B > JiuZhang3.0-8B > JiuZhang3.0-7B.
>
> **W2**. The accuracy of the pre-trained synthetic data raises concerns about the noise introduced by numerous erroneous samples.
> - **Reply to W2**. Actually, it is inevitable that noisy samples are involved into the pre-training data of LLMs, e.g., erroneous and toxic ones[1,2]. Thus, the following SFT process is widely-used to reduce their influence and align LLM with humans. Therefore, after pre-training, we also collect a set of math-related instructions with reliable accuracy for fine-tuning LLMs.
> In our empirical experiments, we have found that such a way can achieve similar performance as filtering the erroneous samples in pre-training data. Concretely, we utilize a well-trained LLM to annotate the accuracy of all the pre-training data, then only select the top-ranked 500k samples for pre-training Mistral-7B. Besides, we also randomly sample 500k samples for pre-training for comparison. After pre-training, we fine-tune the two methods on 50k instructions. As shown in the following table, the two variations achieve comparable performance, indicating that the noise is not an important concern.
>
> [1] Ouyang, Long, et al. "Training language models to follow instructions with human feedback." Advances in neural information processing systems 35 (2022): 27730-27744.
>
> [2] Albalak, Alon, et al. "A survey on data selection for language models." arXiv preprint arXiv:2402.16827 (2024).
>
> |         | GSM8k | MATH |	SVAMP	| ASDiv | MAWPS | CARP | Avg. |
> |------------|------------|------|---------|------------|------------|------|---------|
> | **PT Random 500k + SFT 50k** | 83.2 | 38.4 | 84.9 | 88.3 | 94.8 | 43.6 | 72.2 |
> | **PT Top-Acc. 500k + SFT 50k** | 82.7 | 40.8 | 84.1 | 87.9 | 94.2 | 42.2 | 72.0 |
>
> **W3**. Performance comparison of pre-training using original data versus synthetic data.
> - **Reply to W3**. Following the suggestion of the reviewer, we conduct the experiments that using the original math-related data as the pre-training corpus, and also reuse our SFT data. We train Mistral-7B on the above data, and report the results in the following Table. We can see that the models using our synthetic data perform consistently better than using the original data. It indicates that our synthetic data owns higher quality and is more useful for improving the mathematical reasoning capability.
>
> |         | GSM8k | MATH | SVAMP | ASDiv | MAWPS | CARP |
> |------------|------------|------|---------|------------|------------|------|
> | **Original PT text** | 36.9 | 15.4 | 60.6 | 66.5 | 82.6 | 26.2 |
> | **Original PT text + SFT** | 86.1 | 44.4 | 87.8 | 90.2 | 97.2 | 48.1 |
> | **Our PT text** | 68.7 | 40.2 | 80.4 | 86.5 | 93.5 | 40.2|
> | **Our PT text + SFT** | 88.6 | 52.8 | 90.4 | 92.6 | 97.3 | 51.0|
>
> **W4**. Differences between using gradient descent and semantic similarity methods to filter high-quality data.
> - **Reply to W4**. Following the suggestion of the reviewer, we implement a semantic similarity based methods for data selection. Concretely, we utilize GTE, https://huggingface.co/Alibaba-NLP/gte-large-en-v1.5, a well-performed sentence embedding model  to encode 1M candidates into embeddings, and then compute their semantic similarity with the downstream task data via cosine similarity. Finally, we select top-ranked 100k samples for pre-training, and then fine-tune on 50k instructions. As shown in the following table, our gradient-based method can achieve better performance, indicating its effectiveness for selecting high-quality data.
>
> |         | GSM8k | MATH |	ASDiv | CARP |
> |------------|-----------|--------|------------|------------|
> | **Ours + Embedding**	|78.2|	29.2|	84.8|	31.9|
> | **Ours**	|78.6|	32.8|	84.5|	36.2|
>
> **W5**. Both Table 1 and Table 3 indicate that reasoning with CoT and Tool assistance yields similar performance.
> - **Reply to W5**. Chain-of-thought (CoT) and tool manipulation are two different capabilities for solving reasoning tasks. Existing work mostly focuses on eliciting one of them for 7B LLMs, as it is hard to well master both. As shown in Figure 4 left, the improvement of one of the capabilities, might lead to the degradation of the other one. Therefore, in our work, we aim to ***balance the two capabilities***, and enable our LLM to ***master both well***. To this end, we carefully adjust the proportion of the two kinds of data (see Figure 4 left). Consequently, our LLM can perform better than baselines in the two settings with a balanced performance.

---

> ### Author Response · Authors · 2024-08-04
> **Rebuttal by Authors (part-2)**
>
> We sincerely thank the reviewer for the insightful suggestion and positive feedback, and list our responses for the weaknesses 6-11 below:
>
> **W6**. The loss calculation method for the pre-trained model and its few-shot performance on some benchmarks, like GSM8k and MATH.
> - **Reply to W6**. We follow existing work[3,4] that only computes loss on the tokens in the solution. Following the suggestion of the reviewer, we also test the few-shot performance of our base model (without fine-tuning), and report the results of best-performed baseline DeepSeekMath-base for comparison. As shown in the following table, the few-shot performance of our JiuZhang3.0 also performs better, indicating the effectiveness of our approach.
>
> [3] Yue, Xiang, et al. "Mammoth2: Scaling instructions from the web." arXiv preprint arXiv:2405.03548 (2024).
>
> [4] Cheng, Daixuan, et al. "Instruction Pre-Training: Language Models are Supervised Multitask Learners." arXiv preprint arXiv:2406.14491 (2024).
>
> |         | GSM8k | MATH |	SVAMP	| ASDiv | MAWPS | CARP |
> |------------|------------|------|---------|------------|------------|------|
> | DeepSeekMath-Base	|64.1|	34.2|	73.7|	82.7|	92.7|	44.4 |
> | JiuZhang3.0-7B-Base | 68.7 |40.2	|80.4	|86.5	|93.5	|40.2|
> | JiuZhang3.0-8B-Base | 73.4	|41.2	|83.3	|88.8	|93.4	|38.3|
>
>
> **W7**. Figure 3 illustrates that as the data volume increases, the performance of two base models initially rises and then falls, rather than continuously improving.
> - **Reply to W7**. As shown in Figure 3, we can see that the performance mostly converges well when using 100% data for Mistral-7B, and only falls a little for LLaMA-8B. The reason may be that LLaMA-8B is pre-trained on rather large-scale data (15TB tokens), reducing its data requirement in continual pre-training.
>
> **W8**. Is fine-tuning Llama3 more suitable for your proposed synthetic method compared to Mistral? Which is more appropriate during the SFT phase: Llama3 or Mistral?
> - **Reply to W8**. According to the results in our paper, Mistral-7B with our method can perform better on math problem solving tasks (in Table1), while LLaMA-3 is better on math-related interdisciplinary tasks (in Table2) and tool manipulation tasks (in Table3). The reason may be that Mistral-7B has focused on the math problem solving capability during training, and LLaMA-3 has been pre-trained on more data about interdisciplinary knowledge and code data.
>
> **W9**. Why was 10k selected for training the synthetic model? Could more data be used?
> - **Reply to W9**. As shown in Figure 3 right, the increasing of the training data does not always lead to performance improvement. According to the results, 10k data is sufficient for training the synthetic model, which indicates that data synthesis is not a hard-to-learn capability for small LLMs. Therefore, we only use 10k data for training, which also reduces the cost of using GPT-4 APIs.
>
> **W10**. Comparing the synthetic data capabilities of different LLMs in Table 4 seems unfair, as the study used distilled GPT-4 for synthetic data generation.
> - **Reply to W10**. In Table 4, we aim to study whether small LLMs could be better data synthesis models than other commonly-used larger ones (e.g., ChatGPT, Mixtral-8X7B), for efficiently generating pre-training data. Thus, we focus more on the performance rather than the technical implementation, and maybe the other baselines also have used more high-quality data provided by human or GPT-4. The experimental results have shown that the synthetic data from our model can improve the performance more on the evaluation benchmarks, indicating its effectiveness and verifying our hypothesis.
>
> **W11**. The performance did not significantly improve when comparing different data selection strategies in Table 4.
> - **Reply to W11**. In Table 4, we conduct the experiments on 100k pre-training and 50k fine-tuning samples, due to the limitation on GPU resource. The limited training data causes the performance improvement to be not large, but most of the improvements are significant (with p<0.05) according to our t-test. Besides, with the increasing of the training data scale, the performance gap will become larger. As shown in the following table, when we use 500k pre-training samples and 50k SFT samples, the performance gap between our approach and the variation using random sampling is obvious.
>
> |         | GSM8k | MATH |	ASDiv | CARP |
> |------------|-----------|--------|------------|------------|
> | **Random Sampling PT 100k + SFT 50k**	|78.8 |31.0| 83.4| 34.3|
> | **Ours PT 100k + SFT 50k**	|78.6|	32.8|	84.5|	36.2|
> | **Random Sampling PT 500k + SFT 50k**	|83.2	|38.4|	88.3|	43.6|
> | **Ours PT 500k + SFT 50k**	|85.9	|43.8|	90.0|	45.9|

---

> ### Author Response · Authors · 2024-08-04
> **Rebuttal by Authors (part-3)**
>
> We sincerely thank the reviewer for the insightful suggestion and positive feedback, and list our responses for the weaknesses 12-13 below:
>
> **W12**. The effect of using SFT data on the Mistral-7B base model.
> - **Reply to W12**. Following the suggestion of the reviewer, we directly fine-tune Mistral-7B and LLaMA3-8B on the SFT data (without the synthetic pre-training data), and report the results on the following table. It is obvious that only using the SFT data performs not better than our approach, especially on the more complex MATH and CARP datasets. It indicates the effectiveness of our synthetic pre-training data.
>
> |         | GSM8k | MATH |	SVAMP	| ASDiv | MAWPS | CARP |
> |------------|------------|------|---------|------------|------------|------|
> | **Mistral-7B + SFT only**|87.5|	45.8|	87.9|	89.7	|97|	44.4|
> | **JiuZhang3.0-7B** |	88.6	|52.8|	90.4	|92.6	|97.3|	51|
> |**Llama3-8B + SFT only**|	87.3	|45.8	|86.9	|89.1	|96.8	|44|
> |**JiuZhang3.0-8B**|	88.6	|51	|89.4	|92.6	|97.1	|50.9|
>
> **W13**. There is an error in citation [50].
> - **Reply to W13**. Thank you for your carefully reading. We follow the suggestion in the official website of Cosmopedia https://huggingface.co/datasets/HuggingFaceTB/cosmopedia, and use the given citation format. We will check and refine it in the revised version of our paper.

---

> ### Author Response · Authors · 2024-08-12
> **Urgent Reminder of Rebuttal**
>
> Dear Reviewer 1oiT,
>
> Sorry to disturb you for the last time, but only one day is left until the end of the reviewer-author discussion stage. We still do not know if you have received our response. To address your concerns, we added a number of new experiments to clarify our novelty and contributions, including
> - (1) data noise effect analysis;
> - (2) pre-training using original data;
> - (3) semantic similarity methods to filter data;
> - (4) few-shot performance;
> - (5) different data selection strategies with more data;
> - (6) using SFT data on the Mistral-7B;
>
> You know, conducting the above mentioned experiments within the limited rebuttal period was quite challenging. We would like to know whether our responses have addressed your concerns. If you still have other concerns, please give us an opportunity to clarify them.
>
> Sincerely hope that you can take a moment to reply to us, as it is very important for researchers and their efforts on this work.
>
> Best regards,
>
> The Authors

---

> ### Comment · Reviewer_1oiT · 2024-08-14
>
> Thanks for your detailed response. It addressed some of my concerns. However, Questions 12 does not show the significant improvement of GSM8k, SVAMP, MATH, CARP and other benchmarks after using a large amount of pre-training corpus. The results are not very impressive, so I will doubt the core innovation and effectiveness of this article. I will maintain my current score. And further discuss with other reviewers.

---

> > ### Author Response · Authors · 2024-08-14
> > **New SOTA Results on 23 Datasets by Scaling our Synthetic Data into 10B**
> >
> > Dear Reviewer 1oiT,
> >
> > We sincerely appreciate your thoughtful feedback and your time to review our work. Here, we would like to address your concerns by clarifying the innovation of our approach or reporting new experimental results:
> >
> > 1. Cost-Effectiveness: Our method achieves significant cost reduction compared to baseline works. We use only 4.6B synthesized pre-training tokens, whereas DeepSeekMath uses 120B math corpus, and KP-Math uses 865k training samples generated from GPT-4. As a result, our method achieves better performance on evaluated datasets with nearly 20% of the total cost of the existing state-of-the-art methods. Besides, our method achieves significantly better performance compared to pre-training on the original pre-training corpus for synthesizing data.
> >
> > 2. Scalability: Our method is not only efficient but also highly scalable. When we scaled our training corpus to 10B tokens, we observed further improvements in performance, outperforming baselines in a large margin. We report the results on the following 23 datasets, where the first two table is for natural language reasoning and the last is for tool manipulation settings, respectively. With 10B PT data, our method can lead to great performance gain on Mistral-7B, achieving new SOTA on all the datasets. It demonstrates the potential of our approach to yield even better results with increased data amount.
> >
> > |                                | GSM8k | MATH | SVAMP | ASDiv | MAWPS | CARP |
> > | ------------------------------ | ----- | ---- | ----- | ----- | ----- | ---- |
> > | DeepSeekMath-RL                | 88.2  | 50.2 | 87.3  | 91.8  | 95.5  | **51.6** |
> > | Mistral-7B + 4.6B-PT + SFT     | 88.6  | 52.8 | 90.4  | 92.6  | **97.3**  | 51   |
> > | Mistral-7B + 10B-PT + SFT      | **90.4**  | **56.2** | **91.1**  | **93.3**  | 97.1  | 51.1 |
> >
> >
> > |           | MWPBench/Collge Math                  | MWPBench/GaokaoBench | MWPBench/Fresh Gaokao 2023 | Gaokao2023 Math En | OlympiadBench | DeepMind Mathematics | GSM Plus | Math Odyssey | TheoremQA | Minerva Math |
> > |---------------------------------------|----------------------|----------------------------|--------------------|---------------|----------------------|----------|--------------|-----------|--------------|-------|
> > | DeepSeekMath-7B-RL                   | 37.5                 | 42.7                       | 36.7               | 43.6          | 19.3                 | 57.3     | 66.7         | **36.2**      | **37.9**         | 20.6  |
> > | Mistral 7B + 4.6B PT + SFT                        | 38.2                 | 38.8                       | 43.3               | 48.8          | 22.2                 | 57.2     | 68.4         | 33.9      | 29.3         | 20.2  |
> > | Mistral 7B + 10B PT + SFT             | **41.7**                 | **46.1**                       | **56.7**               | **51.2**          | **25.6**                 | **67.0**     | **71.2**         | 34.4      | 32.5        | **25.4**  |
> >
> > |   | GSM8K                                 | MATH  | GSM Hard | SVAMP   | TabMWP | ASDiv  | MAWPS  |
> > |---------------------------------------|-------|----------|---------|--------|--------|--------|-------|
> > | DeepSeekMath-7b-RL           | 78.5  | 52.8     | 61.3    | 85.9   | **80.3**   | 87.0   | 95.8  |
> > | Mistral 7B + 4.6B PT + SFT                        | 82.4  | 53.0     | **64.9**    | 89.2   | 75.6   | 88.3   | 96.6  |
> > | Mistral 7B + 10B PT + SFT             | **91.1**  | **54.8**     | 61.2    | **91.3**   | 79.6   | **93.4**   | **97.4**  |
> >
> > As we will publicly release all the synthetic data, our method would be a promising solution to reduce the cost of achieving the SOTA performance for LLM. Besides, we believe that we have not even reached the effectiveness ceiling of our method. With more computing resource, it is also promising to further scale up the synthetic dataset or fine-tune stronger LLMs as the data synthetic model.

---

### Author Response · Authors · 2024-08-11
**Kind Reminder of Discussion**

Dear Reviewers,

We sincerely thank all the reviewers for their positive feedback and constructive suggestions. Now, we want to send you a friendly reminder that the discussion stage will be complete soon. We have carefully replied to all the reviewers' concerns and supported our claim with added new experiments and clarification on the rebuttal column. We really want to know whether our responses address your concerns. If there are still any unsolved concerns, please feel free to let us know, and we would be glad to discuss with you.

Thanks,

Authors

---

> ### Comment · Area_Chair_oBmP · 2024-08-13
>
> Thank you for your patience!
> I will notify reviewer 1oiT.

---

> > ### Author Response · Authors · 2024-08-13
> > **Sincerely Thank for AC**
> >
> > Dear AC oBmP,
> >
> > Thank you so much for your attention! After discussion with the three reviewers, we have obtained massive insightful and constructive suggestions. We will revise our paper according to the valuable suggestions!
> >
> > For reviewer 1oiT, we would like to know whether our responses have addressed your concerns. We are very glad to discuss with the reviewer for clarifying the concerns and improving our work.
> >
> > Thank you once again for taking the time to our paper.
> >
> > Best regards,
> >
> > The Authors

---

### Decision · Program_Chairs · 2024-09-25

**Decision:**

Accept (poster)

**Comment:**

This paper proposes a method of enhancing mathematical reasoning in LLMs through cost-efficient data synthesis. It also provides extensive experiments and comprehensive ablation studies to validate the effectiveness of the proposed pipeline, demonstrating superior performance on various mathematical reasoning datasets. It reveals that data synthesis abilities of large models can be effectively distilled into smaller models, enhancing efficiency. Additionally, the paper is pioneering in scaling instruction data over 5M pairs, validating the feasibility and effectiveness of this approach for reasoning tasks.  Reviewers like the proposed cost-efficient method.
At the same time, please follow the promise of releasing the code and data to the research community.